# Quantizable Transformers: Removing Outliers by Helping Attention Heads Do Nothing

**Yelysei Bondarenko, Markus Nagel, Tijmen Blankevoort**
Qualcomm AI Research*
Amsterdam, The Netherlands
{ybond, markusn, tijmen}@qti.qualcomm.com

## Abstract

Transformer models have been widely adopted in various domains over the last years, and especially large language models have advanced the field of AI significantly. Due to their size, the capability of these networks has increased tremendously, but this has come at the cost of a significant increase in necessary compute. Quantization is one of the most effective ways to reduce the computational time and memory consumption of neural networks. Many studies have shown, however, that modern transformer models tend to learn strong outliers in their activations, making them difficult to quantize. To retain acceptable performance, the existence of these outliers requires activations to be in higher bitwidth or the use of different numeric formats, extra fine-tuning, or other workarounds. We show that strong outliers are related to very specific behavior of attention heads that try to learn a "no-op" or just a partial update of the residual. To achieve the exact zeros needed in the attention matrix for a no-update, the input to the softmax is pushed to be larger and larger during training, causing outliers in other parts of the network. Based on these observations, we propose two simple (independent) modifications to the attention mechanism - *clipped softmax* and *gated attention*. We empirically show that models pre-trained using our methods learn significantly smaller outliers while maintaining and sometimes even improving the floating-point task performance. This enables us to quantize transformers to full INT8 quantization of the activations without any additional effort. We demonstrate the effectiveness of our methods on both language models (BERT, OPT) and vision transformers. Our source code is available at https://github.com/qualcomm-ai-research/outlier-free-transformers.

## 1 Introduction

Quantization has been one of the most impactful ways to reduce the computational complexity of transformer networks. Previous work has shown that quantizing networks to 4-bit weights is possible without losing too much accuracy [66, 69]. Some research even shows 4-bit weights might be optimal when trading off model size and bit-width [12].

However, quantizing transformers is not always trivial. When quantizing the activations of a transformer, significant problems arise with outliers in specific layers. This has been noted by several researchers that suggest fixes to transformers after training to ameliorate their effect [13, 67]. These methods are frequently tedious and either require retraining the network, require implementing specific hardware for input-channel quantization [13] or require parts of the activations to still be in higher bit-widths, reducing the effectiveness of the activation quantization [67].

In this paper, we set out to solve the transformer outlier problem entirely by changing the architecture of the network itself. We hope to make transformers easy to quantize from the get-go without needing

---

*Qualcomm AI Research is an initiative of Qualcomm Technologies, Inc.

37th Conference on Neural Information Processing Systems (NeurIPS 2023).

any post-processing. To do so, we thoroughly analyze why these outliers appear. Previous work has found the existence of these outliers [4, 13], but in our work, we come to a fuller understanding of these outlying values. We find that the outliers occur because attention heads are trying not to update the hidden state, and in the process, strong outliers appear due to the softmax function. This happens for language and vision transformers and different specific transformer architectures. This understanding is the foundation for two new tweaks we suggest to transformer architectures that can remove the problem of the outliers entirely.

## 2 Background and related work

In this section, we briefly cover the basics of neural network quantization and discuss why modern transformers are difficult to quantize.

**Quantization** One of the most powerful ways to decrease the computational time and memory consumption of neural networks is quantization, which uses low-bit representations for the weights and activation tensors. On top of that, using low-bit fixed-point representations, such as INT8, one can further reduce energy consumption since the fixed-point operations are more efficient than their floating-point counterparts [23, 59].

We simulate the quantization process in floating-point according to Jacob et al. [26]. We use the following definition of the quantization function:

$$\widehat{\mathbf{x}} := q\left(\mathbf{x}; s, z, b\right) = s \cdot \left(\mathrm{clip}\left(\left\lfloor \frac{\mathbf{x}}{s} \right\rceil + z; 0, 2^b - 1\right) - z\right), \tag{1}$$

where $\mathbf{x}$ denotes the quantizer input (i.e., network weights or activations), $s \in \mathbb{R}_+$ the scale factor or the step-size, $z \in \mathbb{Z}$ the zero point, and $b \in \mathbb{N}$ the bitwidth. $\lfloor \cdot \rceil$ denotes the round-to-nearest-integer operator. This quantization scheme is called *uniform affine* or *asymmetric* quantization [24, 32, 76] and it is one of the most commonly used quantization schemes because it allows for efficient implementation of fixed-point arithmetic. In the case of *symmetric* quantization, we restrict the quantization grid to be symmetric around $z = 0$.

In this work, we focus on *post-training quantization* (PTQ) methods, which take a pre-trained FP32 network and convert it directly into a fixed-point network without the need for the original training pipeline [2, 5, 7, 25, 32, 35, 41, 43, 44, 75]. These methods require either no data or only a small calibration dataset and are easier to use compared to *quantization-aware training* (QAT, Bhalgat et al. 3, Esser et al. 16, Gupta et al. 21, Jacob et al. 26, Krishnamoorthi 32) methods that have you train the entire network for more epochs. For more details on neural network quantization, we refer the reader to [19, 46].

**Outliers in Transformers** Multiple studies have shown that modern transformer-based language models tend to learn outliers in weights and activations [4, 13, 31]. These outliers are present only in a small fixed set of embedding dimensions, but they appear regularly and consistently across multiple layers and data sequences. It was also shown that those outliers play a crucial role in the model predictions and clipping them or by setting to zero the corresponding parameters significantly degrades the model task performance [31, 49]. The strongest in magnitude outliers typically appear at the output of the feed-forward network, FFN, although Dettmers et al. [13] showed that for big enough transformer-based language models they start appearing after every linear layer, including query, key, and value projection layers. This phenomenon holds for many tasks, training objectives and models (both encoder and decoder transformers), including BERT [14], RoBERTa [37], DistilBERT [53], MobileBERT [55], ELECTRA [9], BART [33], XLNet [68], GPT-2 [50], and OPT [74].

Because of these strong outliers, applying per-tensor PTQ for the FFN's output and the residual sum will likely cause a notable error because of the following trade-off between the range and the precision. On the one hand, using a large quantization range for small-ranged values leads to a loss in representation (high rounding error). On the other hand, a small quantization range for large values leads to a very high clipping error. For the case of significant transformer outliers, frequently, no good trade-off can be found between the rounding and clipping error, resulting in an overall high error.

There have been numerous attempts to fix the issue of transformer quantization [4, 12, 13, 17, 27, 28, 51, 54, 62, 63, 69, 71]. Most of these approaches resort to finer quantization granularity (row-wise,

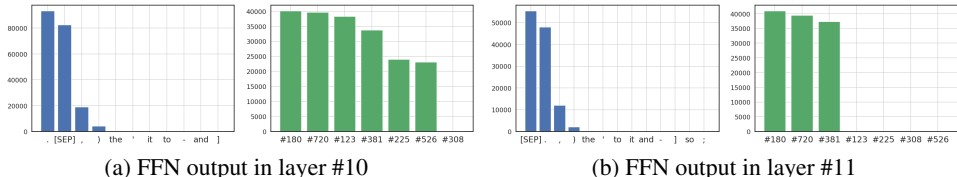

(a) FFN output in layer #10          (b) FFN output in layer #11

Figure 1: Histograms of outlier counts vs. token positions (blue) and hidden dimensions (green), recorded from the MNLI-m validation set on BERT-base. We use zero-based indexing for dimensions.

channel-wise, group-wise weight and activation quantization), use higher bitwidth and/or different numeric format to represent those outliers better or require extra fine-tuning (in the form of QAT and/or knowledge distillation). In other words, they adapt quantization to work with outliers, which often comes at the expense of general applicability or extra inference overhead.

In contrast, in this work, we want to address the root cause of the problem and understand why outliers are learned in the first place and suggest a new pre-training protocol that significantly reduces the magnitude of outliers yielding way more quantization-friendly models that can be effortlessly quantized using PTQ without strong degradation of performance.

## 3   Outlier analysis

**Outliers in BERT models**   In Section 2 we discussed that outliers are present only in a few designated embedding dimensions but they appear regularly and consistently across multiple layers and data sequences. We also discussed that the strongest magnitude outliers in BERT typically appear at the output of FFN in the last encoder layers.

We start by taking the pre-trained *BERT-base-uncased* checkpoint from HuggingFace [65] and fine-tune it on MNLI dataset from the well-known GLUE benchmark [61] (see experimental details in C.1). To identify the outlier dimensions, we pass the MNLI-m validation set through the network and record all outliers[1] at the FFN output in layers #10 and #11[2]. As we can see in Figure 1, there are indeed only a few hidden dimensions where outliers ever occur. We also notice that the majority of outliers ($> 97\%$) correlate with the position of delimiter tokens – [SEP], ".", and ",".

To better understand the role of those outliers, we analyze the attention patterns of the corresponding attention heads. BERT-base uses multi-head attention with $n_{\text{heads}} = 12$ and each head operating on a consecutive subset of $d_{\text{head}} = 64$ features. Therefore, the hidden dimension #180, which happens to have the highest outlier count in both layers #10 and #11, corresponds to attention head #3. In Figure 2 (and more examples in Appendix A.1) we show examples of the attention matrices, values and their product for that head.

A common pattern we found is that the attention head assigns almost all of its probability mass to [SEP] tokens, and other less informative tokens like dots/commas, while these tokens also have small values in $V$ associated with those tokens. This results in a small magnitude product between the two (see Figure 2a). This effectively corresponds to a (soft) *no-update* of the hidden representation, where only small noise is added after the residual. In other cases (Figure 2b and 2c), we observe that a significant portion of attention probability is still spent on delimiter tokens. However, by allocating some of the probability mass on other tokens (together with the small values for the delimiter tokens), this results in a (soft) *selective* update of the hidden representation.

These patterns in self-attention seem to be a learned "workaround" for the limitations of having the softmax and the residual connections in cases where the attention head does not want to update the representation of some or all of the tokens. These observations are in line with Clark et al. [8], Kovaleva et al. [30] that also argued that attending exclusively or almost exclusively to delimiter tokens such as [SEP], periods/commas acts as a "no-op" when the attention head's function is not applicable.

---

[1]We follow Bondarenko et al. [4] and consider outliers as values that exceed 6 standard deviations from the mean of the corresponding activation tensor.

[2]We use 1-based indexing for encoder layers and attention heads throughout the paper.

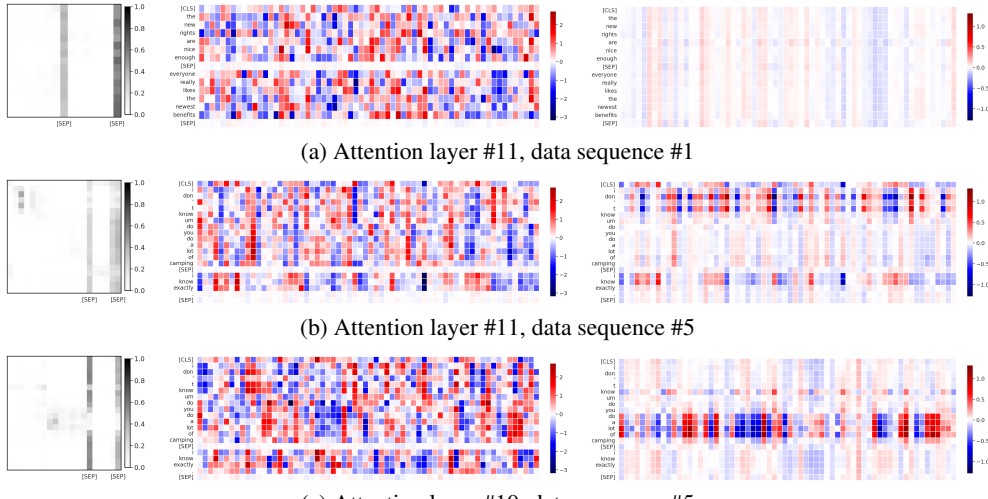

(a) Attention layer #11, data sequence #1

(b) Attention layer #11, data sequence #5

(c) Attention layer #10, data sequence #5

Figure 2: Visualization of the patterns in the self-attention, specifically the attention probabilities, values, and their product (left, middle and right columns, respectively), in attention head #3 for BERT-base, computed on several data sequences from MNLI-m validation set.

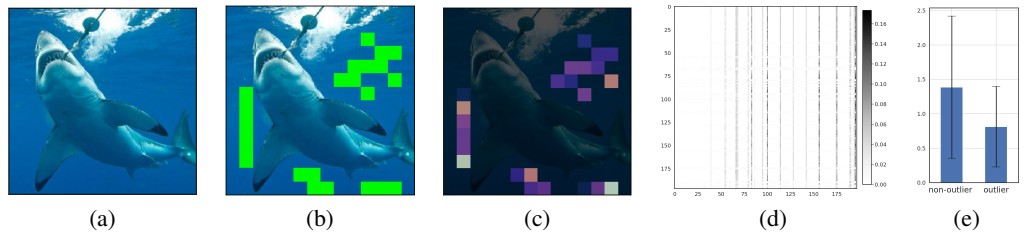

(a)  (b)  (c)  (d)  (e)

Figure 3: A summary of our outlier analysis for ViT demonstrated on a random image from ImageNet validation set. (a) An input image. (b) Outliers in the output of layer #11. (c) Cumulative attention weight spent on every patch (matrix of attention probabilities summed over rows) in the attention head #1, layer #12. (d) A corresponding matrix of attention probabilities. (e) An average magnitude of values for outlier and non-outlier patches.

**Outliers in ViT**   We conduct a similar analysis for Vision transformer [15] trained on ImageNet [52]. For this study, we use a pre-trained checkpoint following our experimental setup from Section 5.

We highlight our findings in Figure 3 and provide more examples in Appendix A.2. Our analysis shows many similarities to the BERT case. Instead of delimiter tokens, the majority of outliers seem to correlate with some random uninformative patches (e.g., in the background). We also see that the corresponding attention head in the next layer allocates the majority of attention probabilities to the same patches. Finally, those outlier patches on average have a distinctly smaller magnitude of values compared to non-outlier ones, leading to similar no-update behavior. The fact that those values are not as close to zero as it was in the BERT case might be related to the smaller model capacity[3], or a relatively shorter training procedure.

**Hypothesis**   Based on these observations, we pose the following hypothesis on how this behavior of attention heads is related to outliers:

1. In order for an attention block to not update a representation of a token on the residual, some attention heads want to allocate most of their attention probability mass to some fixed and common set of tokens that have a low information content (e.g., delimiter tokens or background patches) that can be learned to have a small value function output.

---

[3]We use ViT/S-16 configuration that has only 22M parameters.

2. From the definition of the softmax function[4], it is easy to see that this would require an input of the softmax to have a relatively big dynamic range (Figure 4, 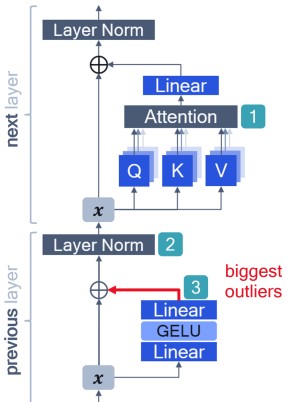①). In fact, in the limit case where softmax is exactly zero, this would require an infinite dynamic range:

$$\text{softmax}(\mathbf{x})_i = 0 \quad \Leftrightarrow \quad \exists j \neq i,\ \mathbf{x}_j - \mathbf{x}_i = +\infty \tag{2}$$

3. Since Layer Normalization ([1], ②) normalizes the outliers, the magnitude of the FFN output *in the previous layer* (③) has to be very high to still produce a sufficiently big dynamic range after the LayerNorm. Note, that this is also applicable for the transformer models with LayerNorm applied prior to the self-attention or linear transformations instead, a variant adopted by GPT, OPT, and many vision transformers [15, 38, 57, 58].

4. Finally, as softmax will never output exact zeros, it will always back-propagate a gradient signal to grow bigger outliers[5]. The outliers will thus tend to become stronger in magnitude, the longer the network is trained.

## 4 Method

In this section, we introduce our proposed modifications for the softmax attention mechanism. Based on our insights from Section 3, the core idea of these modifications is to grant the model the ability to produce very small the magnitude (or even exact zeros) output of attention function, without producing outliers.

Recall that the self-attention [60] is defined as follows:

$$\text{Attention}(\mathbf{x}) := \text{softmax}\left(\frac{\boldsymbol{Q}(\mathbf{x})\boldsymbol{K}(\mathbf{x})^T}{\sqrt{d_{\text{head}}}}\right)\boldsymbol{V}(\mathbf{x}) \tag{3}$$

where $\boldsymbol{Q}$, $\boldsymbol{K}$ and $\boldsymbol{V}$ are learnable linear projections of the input $\mathbf{x}$. Most modern transformer models employ a multi-headed variant of self-attention, where $d_{\text{model}}$ features are partitioned into $n_{\text{heads}}$ groups of $d_{\text{head}}$ features, and the final output is the concatenation of the outputs of (3) applied to each group.

### 4.1 Clipped softmax

First, we propose to replace softmax function in (3) with the following clipped softmax:

$$\text{clipped\_softmax}(\mathbf{x}; \zeta, \gamma) :=$$
$$\text{clip}\left((\zeta - \gamma)\cdot\text{softmax}(\mathbf{x}) + \gamma, 0, 1\right). \tag{4}$$

Here $\mathbf{x}$ is the input and $\zeta \geq 1, \gamma \leq 0$ are the stretch factors which are hyper-parameters of the method. This formulation was proposed before in [40] in the context of binary stochastic gates. We can view (4) as stretching the output of the softmax from $(0,1)$ to $(\gamma, \zeta)$ and then clipping back to $(0,1)$ so that we can represent exact zeros if $\gamma < 0$ and exact ones if $\zeta > 1$. Specifically, the values of the softmax larger than $\frac{1-\gamma}{\zeta-\gamma}$ are rounded to one whereas values smaller than $\frac{-\gamma}{\zeta-\gamma}$ are rounded to zero.

With this drop-in replacement, we can achieve exact zeros (and ones) with a finite range for the softmax input. In addition to that, whenever values are clipped they will not give a gradient, preventing the outliers to grow further.

Figure 4: A schematic illustration of the attention layer in BERT. Hidden activation tensor is denoted by $\mathbf{x}$. $\oplus$ is an element-wise addition. A problematic output of the FFN that generates largest in magnitude outliers is highlighted in red. Notice how those outliers in the *previous layer* influence the behavior in the attention mechanism in the *next layer*.

---

[4] $\text{softmax}(\mathbf{x})_i = \exp(\mathbf{x}_i) / \sum_{j=1}^{d} \exp(\mathbf{x}_j)$
[5] Let $\mathbf{y} = \text{softmax}(\mathbf{x})$. Then $\frac{\partial \mathbf{y}_i}{\partial \mathbf{x}_j} \neq 0\ \forall i, j.$

## 4.2 Gated attention

An alternative way of architecting the model to have a small attention output without outliers is to equip it with an explicit conditional gating mechanism, as shown in Figure 5. The idea is that the model can use the gating to either keep or nullify the update to the representation of certain tokens and not rely on the attention probabilities and values to achieve the same outcome.

Specifically, we propose the following modification to the attention function:

$$\text{Gated\_attention}(\mathbf{x}) := \text{sigmoid}\left(\boldsymbol{G}(\mathbf{x})\right) \odot \text{softmax}\left(\frac{\boldsymbol{Q}(\mathbf{x})\boldsymbol{K}(\mathbf{x})^T}{\sqrt{d_{\text{head}}}}\right)\boldsymbol{V}(\mathbf{x}). \tag{5}$$

Here $\boldsymbol{G}$ is the gating function, $\odot$ is an element-wise multiplication across the token axis and everything else remains the same as in (3). The gating function $\boldsymbol{G}$ is parameterized by a small neural network that is learned jointly with the rest of the model. We replace the attention formulation with the proposed variant in every layer on the transformer network.

**Gating module design** Recall that the input to the attention layer $\mathbf{x}$ has shape $(T, d_{\text{model}})$ that is reshaped into $(n_{\text{heads}}, T, d_{\text{head}})$ for the multi-headed self-attention, where $T$ is the sequence length. We chose to define the gating function on a per-head basis. For each head $i \in \{1, \ldots, n_{\text{heads}}\}$, we specify $\boldsymbol{G}_i : \mathbb{R}^{d_{\text{head}}} \to \mathbb{R}$ and the output of the gating module is $\boldsymbol{\pi}_i \in \mathbb{R}^T$ that is computed as follows:

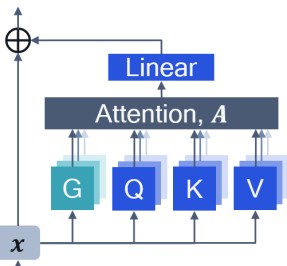

$$\widehat{\boldsymbol{\pi}}_{i,t} = \boldsymbol{G}_i(\mathbf{x}_{i,t,:}) \;\; \forall t \in \{1, \ldots, T\} \tag{6}$$

$$\boldsymbol{\pi}_{i,:} = \text{sigmoid}(\widehat{\boldsymbol{\pi}}_{i,:}), \tag{7}$$

note that gating modules are shared between different token positions but not shared across attention heads.

We want our gating module to be as lightweight as possible. To start with, we experiment with $\boldsymbol{G}_i$'s parameterized by a single linear layer. This gives us a gating module that is computationally inexpensive and has a memory overhead of just $n_{\text{heads}} \cdot (d_{\text{head}} + 1) \sim d_{\text{model}}$ extra parameters (which is equivalent to 1 extra token) per attention layer[6]. We also investigate the effect of using several other gating functions in Appendix B.1.

Figure 5: A schematic illustration of our proposed gated attention.

## 5 Experiments

In this section, we evaluate the proposed modifications to self-attention on several language models (BERT, OPT) and the vision transformers (ViT). We first test the different hyperparameters for the methods and provide insight into how they work. Then we set out to test our method in terms of accuracy, and the difference in quantization improvement after training. All detailed hyperparameters of our experiments are in Appendix C.

**BERT** We experiment with BERT-base-uncased (109M parameters) pre-training using the masked language modeling (MLM) objective. Following [14], we use the concatenation of the training sets of BookCorpus [77] and English Wikipedia[7]. We implement our methods in PyTorch [48] and use training and evaluation pipelines from HuggingFace libraries [20, 34, 65]. We follow closely the pre-training procedure from [14]. To speed up training and experimentation, we train with a maximum sequence length of 128 for the whole duration of the training. We evaluate on Wikipedia validation set and report the MLM perplexity.

**OPT** We experiment with a 125M sized variant of OPT [74] pre-training using the causal language modeling (CLM) objective. Due to compute constraints, we train the model on the same dataset that was used for BERT pre-training (BookCorpus + Wikipedia) with a maximum sequence length of 512

---

[6]For instance, in case of BERT-base, this amounts to less than 0.009% of the total model size.

[7]Specifically, we use the English subset of Wiki-40b, `https://huggingface.co/datasets/wiki40b`, that contains cleaned-up text of English Wikipedia and training/validation splits.

| $\gamma$ | $\zeta$ | FP16 ppl.↓ | Max inf. norm | Avg. kurtosis | W8A8 ppl.↓ |
|---|---|---|---|---|---|
| 0 (= Vanilla) | 1 | $4.49^{\pm 0.01}$ | $735^{\pm 55}$ | $3076^{\pm 262}$ | $1294^{\pm 1046}$ |
| 0 | 1.003 | $4.48^{\pm 0.01}$ | $715^{\pm 335}$ | $2159^{\pm 238}$ | $451^{\pm 57}$ |
| 0 | 1.03 | $4.49^{\pm 0.00}$ | $741^{\pm 66}$ | $1707^{\pm 1249}$ | $1469^{\pm 646}$ |
| $-0.003$ | 1 | $4.46^{\pm 0.00}$ | $688^{\pm 64}$ | $2149^{\pm 110}$ | $636^{\pm 566}$ |
| $-0.03$ | 1 | $\mathbf{4.41^{\pm 0.01}}$ | $\mathbf{20^{\pm 1}}$ | $\mathbf{80^{\pm 6}}$ | $\mathbf{4.55^{\pm 0.01}}$ |
| $-0.003$ | 1.003 | $4.47^{\pm 0.00}$ | $683^{\pm 23}$ | $2494^{\pm 1205}$ | $268^{\pm 120}$ |
| $-0.03$ | 1.03 | $\mathbf{4.43^{\pm 0.03}}$ | $\mathbf{22^{\pm 3}}$ | $\mathbf{73^{\pm 8}}$ | $\mathbf{4.56^{\pm 0.05}}$ |

Table 1: The impact of clipped softmax hyperparameters on BERT-base.

and batch size of 192. Similar to our BERT experiments, we use training and evaluation pipelines from HuggingFace libraries. We evaluate on Wikipedia validation set and report the CLM perplexity.

**ViT**  Finally, we explore the effectiveness of proposed techniques on vision transformer [15] (*ViT-S/16* configuration, 22M parameters) trained on ImageNet-1K [11, 52]. For these experiments, we adopt the training and validation pipelines from PyTorch Image models library [64]. We report top-1 accuracy on the validation set of ImageNet.

**Quantization setup**  In all experiments, after the model is trained, we apply 8-bit PTQ. We use uniform affine quantization – symmetric weights, asymmetric activations – with the static activation range setting, as discussed in Section 2. We quantize all weights and activations (both input and output), except the final linear layer for BERT and OPT models. We explore several choices of range estimation (see Appendix C.4) and report the best configuration for each experiment, based on the model performance. We repeat each PTQ experiment 3 times with different random seeds[8] and report mean and standard deviation for accuracy/perplexity.

We train each network two times with different random seeds and report mean and standard deviation. To assess the amount of outliers in the trained model, we use two metrics: the maximum $\|\mathbf{x}\|_\infty$ averaged across the validation set, and *kurtosis* of $\mathbf{x}$ averaged across all layers, where $\mathbf{x}$ is the output of an attention layer. These metrics have been shown to correlate well with the model quantizability [4, 6].

### 5.1   The impact of clipped softmax hyperparameters ($\gamma$ and $\zeta$)

We investigate the effect of different values of the clipped softmax stretch parameters and present the results in Table 1. We can see that most of the improvement happens when we use $\gamma < 0$ (clipping at zero). For instance, using the value of $\gamma = -0.03$ leads to a significantly smaller infinity norm, kurtosis, and quantized model perplexity, compared to the baseline. It is also clear that in the limit $|\gamma| \to 0$ we approach the vanilla softmax attention. Using $\zeta > 1$ (clipping at one) yields similar results to the vanilla softmax. Finally, when we combine both $\gamma < 0$ and $\zeta > 1$, for which the results seem similar to just clipping at 0. We, therefore, conclude that for dampening outliers, only the lower-range clipping allows exact zeros matter. Going forward we use only $\gamma < 0$ and in Appendix B.5 we confirm that $\zeta > 1$ is not required for ViT.

These observations are in line with our hypothesis that by giving the model the mechanism for representing exact zeros in the attention, we don't need to learn the strong outliers.

### 5.2   Clipped softmax $\gamma$ vs. sequence length

As having an extra hyper-parameter that needs to be tuned per model or setup is generally not desirable, we study the sensitivity of the stretch factor $\gamma$ and its relation with the sequence length $T$. Recall that the matrix of attention probabilities $\boldsymbol{P}$ has dimensions $T \times T$ and each row sums up to one. Because of that, the average value in $\boldsymbol{P}$ is $1/T$. It is reasonable to assume that if we define

---

[8]Different random subsets of training data are used for quantizer range estimation.

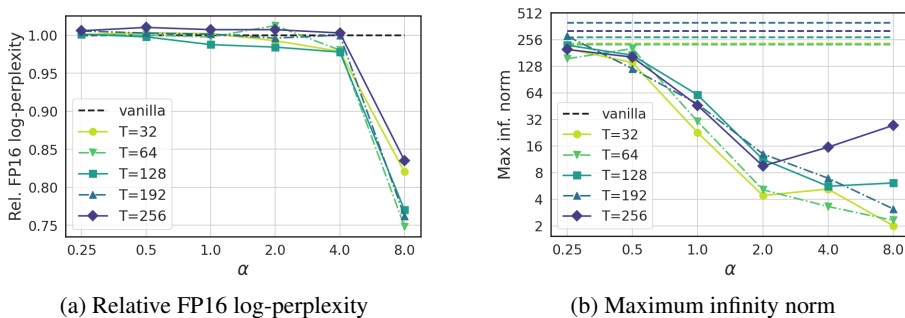

(a) Relative FP16 log-perplexity

(b) Maximum infinity norm

Figure 6: The performance of clipped softmax using $\gamma = -\alpha/T$ parameterization on BERT-6L. (a) Relative (compared to vanilla softmax pre-training) FP16 log-perplexity ↑ on Wikitext validation set. (b) Maximum infinity norm of the attention layer output (note the logarithmic y-axis).

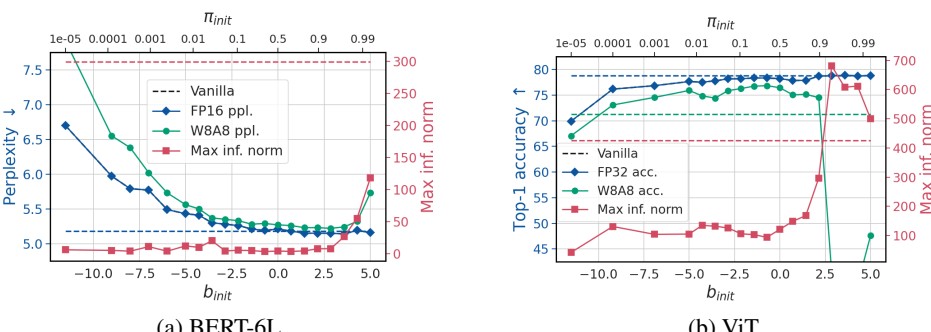

(a) BERT-6L

(b) ViT

Figure 7: The performance of Linear gated attention using different bias initialization settings.

$\gamma := -\frac{\alpha}{T}$, where $\alpha > 0$ is a new hyperparameter, there might be a set or a range of values of $\alpha$ that works well across different sequence lengths.

To study this, we train a 6-layer variant of BERT-base (BERT-6L) for 500000 steps on WikiText-103 [42] with a batch size of 128 with several values of maximum sequence lengths $T \in \{32, 64, 128, 192, 256\}$ and values of $\alpha \in \{1/4, 1/2, 1, 2, 4, 8\}$. As we can see from Figure 6, using a clipped softmax with $\alpha \in [2, 4]$ significantly dampens the magnitude of outliers while maintaining good FP16 perplexity across all explored sequence lengths.

### 5.3 The impact of bias initialization in gated attention

In all our gated attention experiments, we randomly initialize the weights of $\boldsymbol{G}$, following [22]. By initializing the *bias* to a specific value, however, we can set gates to be more *open* or more *closed* initially. More open at the start means we initialize closer to the original network, but given the exponential nature of the gate it might take many iterations for the gate to learn to close. Similarly, if the gates are all closed at the start, we deviate too far from the original model training, causing a potential decrease in performance. Assuming Linear $\boldsymbol{G}_i$'s with small initial weights, if we set the bias to the value of $b_{\text{init}}$, then $\boldsymbol{G}_i(\cdot) \approx b_{\text{init}}$ and $\boldsymbol{\pi}_i(\cdot) = \text{sigmoid}(\boldsymbol{G}_i(\cdot)) \approx \text{sigmoid}(b_{\text{init}}) =: \pi_{\text{init}}$, at the start of training.

We study the effect of different values of $b_{\text{init}}$ for Linear gated attention on BERT-6L and ViT. We set the bias for all $\boldsymbol{G}_i$'s to the same value of $b_{\text{init}}$. For BERT-6L, we use the same setup as in Section 5.2, with a fixed sequence length of 128. For ViT, we use the main setup, except we train it for 150 epochs instead of 300.

In Figure 7 we see in both BERT and ViT cases that using bias with very high $\pi_{\text{init}}$ generally performs similarly to the vanilla attention (comparable floating-point performance but strong outliers and poor quantized performance) while setting bias to have very low $\pi_{\text{init}}$ dampens outliers quite well but leads to strong degradation in the floating-point and quantized performance. The reasonable ranges of $\pi_{\text{init}}$ seems to be around $[0.25, 0.9]$ for BERT and $[0.1, 0.5]$ for ViT. The wide range indicates the relative robustness of our method to this hyperparameter.

| Model | Method | FP16/32 | Max inf. norm | Avg. kurtosis | W8A8 |
|---|---|---|---|---|---|
| BERT (ppl.↓) | Vanilla | $4.49^{\pm0.01}$ | $735^{\pm55}$ | $3076^{\pm262}$ | $1294^{\pm1046}$ |
| | Clipped softmax | $\mathbf{4.39}^{\pm\mathbf{0.00}}$ | $\mathbf{21.5}^{\pm\mathbf{1.5}}$ | $\mathbf{80}^{\pm\mathbf{6}}$ | $\mathbf{4.52}^{\pm\mathbf{0.01}}$ |
| | Gated attention | $4.45^{\pm0.03}$ | $39.2^{\pm26.0}$ | $201^{\pm181}$ | $4.65^{\pm0.04}$ |
| OPT (ppl.↓) | Vanilla | $15.84^{\pm0.05}$ | $340^{\pm47}$ | $1778^{\pm444}$ | $21.18^{\pm1.89}$ |
| | Clipped softmax | $16.29^{\pm0.07}$ | $63.2^{\pm8.8}$ | $19728^{\pm7480}$ | $37.20^{\pm2.40}$ |
| | Gated attention | $\mathbf{15.55}^{\pm\mathbf{0.05}}$ | $\mathbf{8.7}^{\pm\mathbf{0.6}}$ | $\mathbf{18.9}^{\pm\mathbf{0.9}}$ | $\mathbf{16.02}^{\pm\mathbf{0.07}}$ |
| ViT (acc.↑) | Vanilla | $80.75^{\pm0.10}$ | $359^{\pm81}$ | $1018^{\pm471}$ | $69.24^{\pm6.93}$ |
| | Clipped softmax | $80.89^{\pm0.13}$ | $\mathbf{73.7}^{\pm\mathbf{14.9}}$ | $22.9^{\pm1.6}$ | $79.77^{\pm0.25}$ |
| | Gated attention | $\mathbf{81.01}^{\pm\mathbf{0.06}}$ | $79.8^{\pm0.5}$ | $\mathbf{19.9}^{\pm\mathbf{0.3}}$ | $\mathbf{79.82}^{\pm\mathbf{0.11}}$ |

Table 2: A summary of results for our proposed methods applied on BERT, OPT-125m, and ViT.

| Model | Method | FP16 | Max inf. norm | Avg. kurtosis | W8A8 |
|---|---|---|---|---|---|
| OPT-350m (ppl.↓) | Vanilla | 13.19 | 253 | 2689 | $37.52^{\pm3.84}$ |
| | Gated attention | 13.01 | 65.4 | 261 | $\mathbf{14.42}^{\pm\mathbf{0.06}}$ |
| OPT-1.3B (ppl.↓) | Vanilla | 12.13 | 428 | 2756 | $989.6^{\pm175}$ |
| | Gated attention | 12.21 | 67.2 | 444 | $\mathbf{29.95}^{\pm\mathbf{0.42}}$ |

Table 3: The performance of gated attention applied on bigger variants of OPT model.

## 5.4 Main results

We summarize our main set of results in Table 2. As we can see, in almost all cases, both of our proposed techniques dampen the outliers' magnitude to a great extent, reduce the kurtosis, and yield models with significantly higher quantized performance, which is close to the original FP16/32 performance. In addition to that, for each model, at least one of our methods also improves the floating-point task performance. We hypothesize this is because the network is helped with learning the "no-op" updates more easily. However, we are cautious about the improved performance as this is not consistent across all hyper-parameters and it is unclear if it generalizes to more architectures and larger models.

The only case where our method failed to perform well was the clipped softmax applied to OPT. At the moment, we do not have an explanation of why this is the case and leave it for future work. We list selected hyper-parameters and show extended results in Appendix B. We also show the results of our proposed methods quantized to lower bitwidths in Appendix B.7.

**Results for bigger models** We study the question of scalability of our methods to larger models. In Table 3 we show the gated attention results for 350m and 1.3B variants of OPT. Due to compute constraints, we trained networks for $10^5$ steps with batch size of 256 and the rest is the same as in our main pre-training setup. As we can see, our proposed gated attention is also very effective at dampening the outliers and significantly improving the quantized model performance when applied to bigger models. We further study in Appendix B.6 how gated attention can decrease outliers when fine-tuning bigger pre-trained models with outliers.

## 5.5 Qualitative results

In Figure 8 we compare the learned attention patterns using vanilla softmax and our proposed methods (more examples in Appendix A.1). As we can see, both methods can represent a partial/soft no-op behavior, but in case of our methods this does not require strong outliers elsewhere in the network. Note that we found similar patterns in multiple attention heads, but the exact head indices where we observed such patterns depend on random initialization. In the case of clipped softmax, the attention probabilities are generally more diffused and smaller in magnitude (which comes from the stretching and clipping). In the case of gated attention, the output of the softmax is significantly different since the update of the hidden representation is now further modulated by gating probabilities.

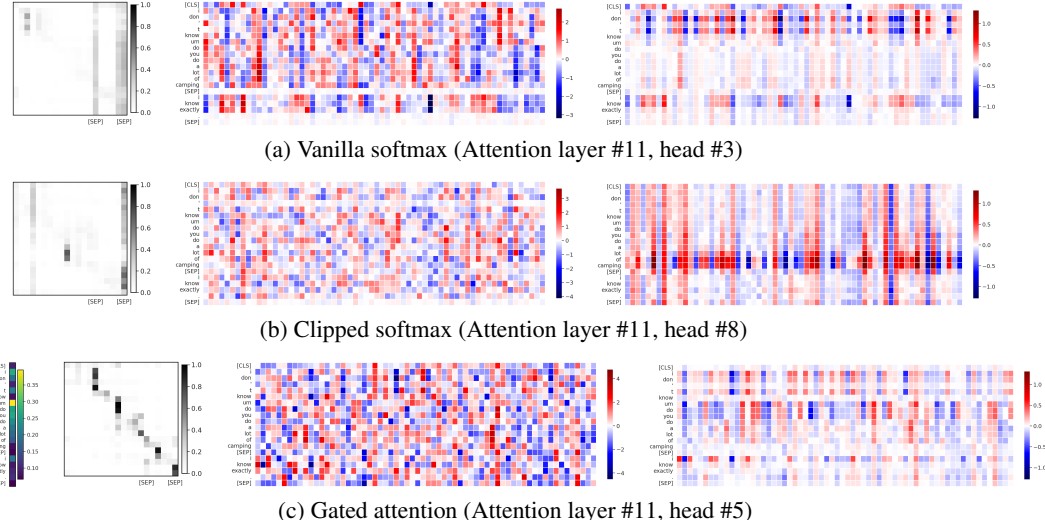

(a) Vanilla softmax (Attention layer #11, head #3)

(b) Clipped softmax (Attention layer #11, head #8)

(c) Gated attention (Attention layer #11, head #5)

Figure 8: Visualization of the self-attention patterns for BERT-base trained using vanilla and our proposed techniques, computed on data sequence #5 from MNLI-m validation set. (a), (b): attention probabilities, values, and their product. (c): gating probabilities $\pi = \text{sigmoid}\left(\boldsymbol{G}\left(\mathbf{x}\right)\right)$, attention probabilities (output of softmax), values, and their combined product.

# 6 Discussion

**"No-op" behavior**  It is interesting to note that the identified "no-op" behavior is likely not limited to transformers and that convolutional architectures likely learn something similar. We also see that despite the network trying to learn a full "no-op", still a small amount of noise is added to each residual, which may constitute a form of network regularization. Investigating this further might give us a clue as to why neural networks generalize despite being significantly overparametrized if many parameters are rendered unused by not updating the representation in later layers [72].

**Limitations**  We have not studied the effect of our method on large-scale transformers, as it would require training very expensive models from scratch. Given the fundamental understanding of the issue underlying our solutions, we expect the same effect on large-scale models. We show a very small improvement in FP16/FP32 performance due to our methods, but we do not deem our results exhaustive enough to claim that this will hold in general. Lastly, our methods do have a hyperparameter each, although we show that both methods are relatively robust to its hyperparameter, having one is never optimal.

**Impact**  As our methods help transformers to be more efficient, we expect only positive outcomes of our work. Making neural networks more efficient will help with their high power consumption at inference. It further helps to move inference from the cloud to edge devices which can overcome potential privacy concerns. We cannot fathom any negative impact from our work that is not severely construed.

# 7 Conclusions

We have thoroughly analyzed the activation outlier problem that makes transformers difficult to quantize. We showed that transformer networks try to learn not to update residuals and that by doing so, through the combination of the softmax, residual connections and LayerNorm, significant outliers appear in transformers. Based on this insight, we proposed two methods to address this at the core – *clipped softmax* and *gated attention*. These structural changes to transformers give similar, if not better, floating-point performance after training but significantly improve the post-training quantization results. We hope that with these two architectural changes to transformers, anyone can train high-performance transformers that are easy to quantize and can benefit from efficient integer inference.

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

# Supplementary materials

## A  Additional graphs from outlier analysis

In this section, we present additional graphs from our outlier investigation in Section 3 for BERT and vision transformer.

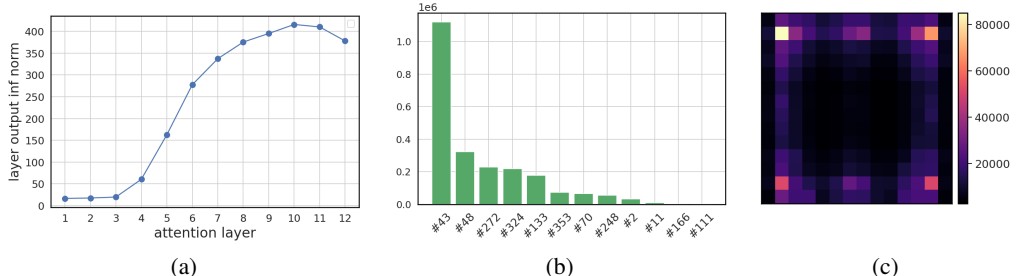

|(a)|(b)|(c)|

Figure 9: A summary of several outlier statistics recorded from ImageNet validation set on ViT. (a) Average infinity norm of the output of each attention layer. (b) A histogram of outlier counts in attention layer #10 vs. hidden dimensions. We use zero-based indexing for dimensions. (c) A heatmap of outlier counts in attention layer #10 vs. patch positions.

### A.1  BERT

Recall from Figure 1 that all the outliers are only present in hidden dimensions #123, #180, #225, #308, #381, #526, #720 (with the majority of them in #180, #720). These hidden dimensions correspond to attention heads #2, #3, #4, #5, #6, #9, and #12. In Figures 10 and 11 we show more examples of the discovered self-attention patterns for attention heads #3 and #12 ($\leftrightarrow$ hidden dim #180 and #720, respectively). We also show self-attention patterns in attention heads and layers which are not associated with the outliers in Figures 12 and 13, respectively. Finally, in Figures 14 and 15 we show more examples of the attention patterns learned in the network trained with clipped softmax and gated attention.

### A.2  ViT

Figure 9 further shows that there are a lot of similarities in the outlier behavior in the vision transformer, compared to BERT. The strongest magnitude outliers generally happen in the later layers, peaking at layers #10 and #11. The majority of outliers ($> 99\%$) are only ever happening in only 10 hidden dimensions, primarily in dimensions #48 and #43, which corresponds to the attention head #1. Finally, averaged across the entire ImageNet validation set, the outliers seem to be concentrated at the boundaries of the image, which suggest a strong correlation with the background (and a negative correlation with the object, which is usually in the center of the image in the ImageNet dataset).

In Figures 16 and 17, we show more examples of outlier and self-attention patterns in the attention head #1 ($\leftrightarrow$ hidden dimensions #48, #43) for a random subset of images from the ImageNet validation set (in layers #10 and #11, respecively).

## B  Detailed results

In this section, we provide extended results for each model, including the used hyperparameters and other design choices. We also present some additional ablation studies.

| Configuration | $G$ | Memory overhead (per attention layer) | |
| --- | --- | --- | --- |
| | | # extra parameters | # extra tokens |
| Linear | $n_{\text{heads}} \times \text{Linear}(d_{\text{head}} \rightarrow 1)$ | $n_{\text{heads}}(d_{\text{head}} + 1)$ | $\sim 1$ |
| MLP | $n_{\text{heads}} \times \text{MLP}(d_{\text{head}} \rightarrow n_{\text{hid}} \rightarrow 1)$ | $n_{\text{heads}}(n_{\text{hid}}(d_{\text{head}} + 2) + 1)$ | $\sim n_{\text{hid}}$ |
| All-heads-linear | $\text{Linear}(d_{\text{model}} \rightarrow n_{\text{heads}})$ | $n_{\text{heads}}(d_{\text{model}} + 1)$ | $\sim n_{\text{heads}}$ |

Table 4: An overview of the gating function parameterizations explored in this paper and their memory overhead.

## B.1 Gating architectures

We investigate the choice of several gating functions, summarized in Table 4. The configuration "MLP" parameterizes each $G_i$ with a feed-forward net with one hidden layer of size $n_{\text{hid}}$ and a ReLU non-linearity [47]. We also explore what happens if we allow the mixing of the representation from different attention heads in the "All-heads-linear" setting, where we use a single linear layer to produce the gating probabilities for all attention heads at once. All three options are tested below. Unless explicitly stated otherwise, we initialize the bias of the gating function to zero (i.e., $b_{\text{init}} = 0$ $\leftrightarrow \pi_{\text{init}} = 0.5$).

## B.2 BERT

| Method | FP16 ppl.$\downarrow$ | Max inf norm | Avg. Kurtosis | W8A8 ppl.$\downarrow$ |
| --- | --- | --- | --- | --- |
| Vanilla | $4.49^{\pm 0.01}$ | $735.0^{\pm 54.9}$ | $3076^{\pm 262}$ | $1294^{\pm 1046}$ |
| CS ($\gamma = -0.005$) | $4.44^{\pm 0.02}$ | $406.6^{\pm 35.2}$ | $1963^{\pm 753}$ | $75.27^{\pm 39.57}$ |
| CS ($\gamma = -0.01$) | $4.35^{\pm 0.01}$ | $198.3^{\pm 78.7}$ | $1581^{\pm 839}$ | $7.06^{\pm 2.37}$ |
| CS ($\gamma = -0.015$) | $4.37^{\pm 0.01}$ | $38.9^{\pm 7.9}$ | $165^{\pm 34}$ | $4.54^{\pm 0.01}$ |
| CS ($\gamma = -0.02$) | $4.39^{\pm 0.02}$ | $31.7^{\pm 6.3}$ | $90^{\pm 20}$ | $4.56^{\pm 0.02}$ |
| CS ($\gamma = -0.025$) | $\mathbf{4.39^{\pm 0.00}}$ | $\mathbf{21.5^{\pm 1.5}}$ | $\mathbf{80^{\pm 6}}$ | $\mathbf{4.52^{\pm 0.01}}$ |
| CS ($\gamma = -0.03$) | $4.41^{\pm 0.01}$ | $20.4^{\pm 0.2}$ | $79^{\pm 6}$ | $4.55^{\pm 0.01}$ |
| CS ($\gamma = -0.04$) | $4.51^{\pm 0.05}$ | $19.8^{\pm 9.0}$ | $85^{\pm 7}$ | $4.65^{\pm 0.06}$ |
| GA, Linear ($\pi_{\text{init}} = 0.25$) | $4.49^{\pm 0.00}$ | $139.8^{\pm 62.3}$ | $739^{\pm 412}$ | $5.05^{\pm 0.27}$ |
| GA, Linear ($\pi_{\text{init}} = 0.5$) | $4.48^{\pm 0.00}$ | $177.3^{\pm 33.2}$ | $652^{\pm 81}$ | $5.13^{\pm 0.15}$ |
| GA, Linear ($\pi_{\text{init}} = 0.75$) | $4.49^{\pm 0.00}$ | $71.4^{\pm 49.9}$ | $262^{\pm 147}$ | $4.88^{\pm 0.22}$ |
| GA, Linear ($\pi_{\text{init}} = 0.9$) | $4.49^{\pm 0.00}$ | $171.5^{\pm 8.8}$ | $559^{\pm 141}$ | $5.15^{\pm 0.03}$ |
| GA, MLP ($n_{\text{hid}} = 4$) | $\mathbf{4.45^{\pm 0.03}}$ | $\mathbf{39.2^{\pm 26.0}}$ | $\mathbf{201^{\pm 181}}$ | $\mathbf{4.65^{\pm 0.04}}$ |
| GA, MLP ($n_{\text{hid}} = 64$) | $4.49^{\pm 0.01}$ | $117.0^{\pm 48.3}$ | $507^{\pm 167}$ | $4.77^{\pm 0.01}$ |
| GA, All-heads-linear | $4.49^{\pm 0.01}$ | $58.3^{\pm 41.2}$ | $334^{\pm 321}$ | $4.67^{\pm 0.03}$ |

Table 5: Main results for our proposed clipped softmax (CS) and gated attention (GA) applied to BERT-base. We report the masked language modeling perplexity (ppl. for short) on the English Wikipedia validation set for both the floating-point baseline and W8A8 quantized model. We also report the maximum $\|\mathbf{x}\|_\infty$ averaged across the validation set, and kurtosis of $\mathbf{x}$ averaged across all layers, where $\mathbf{x}$ is the output of an attention layer.

Detailed results for BERT-base are summarized in Table 5. As we can see, across most of the settings, both of our methods significantly dampen the outliers' magnitude, reduce the kurtosis, drastically improve the quantized performance, while maintaining and sometimes improving the FP16 perplexity.

## B.3 OPT

Detailed results for OPT-125m are summarized in Table 6.

In our early experiments on a smaller OPT model, we found that applying the weight decay on LayerNorm weights $\boldsymbol{\gamma}$ (which isn't the case, by default) has a strong effect on reducing the outliers' magnitude while yielding the comparable FP16 performance. Therefore, we present the results of applying our gated attention approach in both cases, with and without applying weight decay on LN $\boldsymbol{\gamma}$. As we can see in Table 6, in both cases gated attention (further) dampens the outliers' magnitude to a

| Method | LN $\gamma$ wd | FP16 ppl.↓ | Max inf norm | Avg. Kurtosis | W8A8 ppl.↓ |
|---|---|---|---|---|---|
| Vanilla | ✗ | $15.84^{\pm0.05}$ | $339.6^{\pm47.2}$ | $1777^{\pm444.}$ | $21.18^{\pm1.89}$ |
| GA, Linear ($\pi_{\text{init}} = 0.1$) | ✗ | $15.61^{\pm0.05}$ | $35.6^{\pm4.5}$ | $42.4^{\pm22.9}$ | $16.41^{\pm0.18}$ |
| GA, Linear ($\pi_{\text{init}} = 0.25$) | ✗ | $\mathbf{15.50^{\pm0.04}}$ | $\mathbf{35.8^{\pm0.5}}$ | $\mathbf{59.0^{\pm48.3}}$ | $\mathbf{16.25^{\pm0.08}}$ |
| GA, Linear ($\pi_{\text{init}} = 0.5$) | ✗ | $15.54^{\pm0.01}$ | $46.5^{\pm5.0}$ | $40.6^{\pm8.9}$ | $16.30^{\pm0.01}$ |
| GA, All-heads-linear | ✗ | $15.43^{\pm0.01}$ | $32.8^{\pm1.7}$ | $24.2^{\pm3}$ | $16.30^{\pm0.12}$ |
| Vanilla | ✓ | $15.96^{\pm0.03}$ | $87.7^{\pm31.9}$ | $2080^{\pm1460}$ | $39.46^{\pm16.59}$ |
| CS ($\gamma = -1/512$) | ✓ | $15.99^{\pm0.02}$ | $106.4^{\pm7.0}$ | $5764^{\pm2150}$ | $185.23^{\pm220.00}$ |
| CS ($\gamma = -2/512$) | ✓ | $15.90^{\pm0.02}$ | $102.0^{\pm27.0}$ | $11290^{\pm4372}$ | $60.90^{\pm52.70}$ |
| CS ($\gamma = -4/512$) | ✓ | $15.86^{\pm0.01}$ | $83.1^{\pm20.6}$ | $17174^{\pm7791}$ | $84.64^{\pm10.55}$ |
| CS ($\gamma = -8/512$) | ✓ | $16.13^{\pm0.09}$ | $61.5^{\pm9.9}$ | $19204^{\pm4284}$ | $42.62^{\pm3.64}$ |
| CS ($\gamma = -12/512$) | ✓ | $16.29^{\pm0.07}$ | $63.2^{\pm8.8}$ | $19727^{\pm7479}$ | $37.22^{\pm2.39}$ |
| GA, Linear ($\pi_{\text{init}} = 0.1$) | ✓ | $15.69^{\pm0.05}$ | $7.3^{\pm0.4}$ | $25.4^{\pm10}$ | $16.23^{\pm0.08}$ |
| GA, Linear ($\pi_{\text{init}} = 0.25$) | ✓ | $\mathbf{15.55^{\pm0.05}}$ | $\mathbf{8.7^{\pm0.6}}$ | $\mathbf{18.9^{\pm1}}$ | $\mathbf{16.02^{\pm0.07}}$ |
| GA, Linear ($\pi_{\text{init}} = 0.5$) | ✓ | $15.63^{\pm0.00}$ | $10.8^{\pm0.7}$ | $42.0^{\pm19}$ | $16.20^{\pm0.01}$ |
| GA, All-heads-linear | ✓ | $15.53^{\pm0.01}$ | $7.9^{\pm0.3}$ | $13.8^{\pm1}$ | $16.09^{\pm0.08}$ |

Table 6: Main results for our proposed clipped softmax (CS) and gated attention (GA) applied to OPT-125m. We report the causal language modeling perplexity (ppl. for short) on the English Wikipedia validation set for both the floating-point baseline and W8A8 quantized model. We also report the maximum $\|\mathbf{x}\|_\infty$ averaged across the validation set, and kurtosis of $\mathbf{x}$ averaged across all layers, where $\mathbf{x}$ is the output of an attention layer.

great extent, reduces the kurtosis, and yields models with significantly higher quantized performance, which is close to the original FP16 performance.

## B.4 ViT

| Method | Patch. Embd. LN | FP32 acc. | Max inf norm | Avg. Kurtosis | W8A8 acc. |
|---|---|---|---|---|---|
| Vanilla | ✗ | $80.75^{\pm0.10}$ | $358.5^{\pm81.2}$ | $1018.3^{\pm471.5}$ | $69.24^{\pm6.93}$ |
| CS ($\gamma = -0.003$) | ✗ | $80.24^{\pm0.05}$ | $69.3^{\pm20.7}$ | $25.6^{\pm8.6}$ | $78.71^{\pm0.33}$ |
| CS ($\gamma = -0.004$) | ✗ | $80.38^{\pm0.01}$ | $74.9^{\pm10.6}$ | $30.6^{\pm4.9}$ | $78.66^{\pm0.49}$ |
| GA, Linear ($\pi_{\text{init}} = 0.25$) | ✗ | $80.62^{\pm0.01}$ | $86.0^{\pm8.0}$ | $23.4^{\pm2.7}$ | $79.16^{\pm0.05}$ |
| GA, Linear ($\pi_{\text{init}} = 0.5$) | ✗ | $80.32^{\pm0.02}$ | $88.4^{\pm17.9}$ | $27.9^{\pm14.0}$ | $78.90^{\pm0.25}$ |
| GA, MLP ($n_{\text{hid}} = 4$) | ✗ | $80.62^{\pm0.05}$ | $118.2^{\pm40.5}$ | $47.8^{\pm29.8}$ | $78.79^{\pm0.29}$ |
| Vanilla | ✓ | $80.98^{\pm0.08}$ | $81.1^{\pm2.5}$ | $24.5^{\pm1.8}$ | $79.62^{\pm0.06}$ |
| CS ($\gamma = -0.0001$) | ✓ | $\mathbf{80.89^{\pm0.13}}$ | $\mathbf{73.7^{\pm14.9}}$ | $\mathbf{22.9^{\pm1.6}}$ | $\mathbf{79.77^{\pm0.25}}$ |
| CS ($\gamma = -0.0003$) | ✓ | $80.92^{\pm0.07}$ | $78.9^{\pm5.5}$ | $23.8^{\pm0.5}$ | $79.63^{\pm0.05}$ |
| CS ($\gamma = -0.0005$) | ✓ | $80.95^{\pm0.08}$ | $72.9^{\pm11.8}$ | $24.4^{\pm0.7}$ | $79.73^{\pm0.08}$ |
| CS ($\gamma = -0.001$) | ✓ | $80.95^{\pm0.16}$ | $80.8^{\pm2.1}$ | $24.1^{\pm0.7}$ | $79.69^{\pm0.03}$ |
| CS ($\gamma = -0.002$) | ✓ | $80.80^{\pm0.07}$ | $78.0^{\pm0.5}$ | $25.8^{\pm0.7}$ | $79.32^{\pm0.07}$ |
| CS ($\gamma = -0.003$) | ✓ | $80.79^{\pm0.02}$ | $75.6^{\pm7.9}$ | $28.1^{\pm4.0}$ | $79.00^{\pm0.10}$ |
| GA, Linear ($\pi_{\text{init}} = 0.5$) | ✓ | $\mathbf{81.01^{\pm0.06}}$ | $\mathbf{79.8^{\pm0.5}}$ | $\mathbf{19.9^{\pm0.3}}$ | $\mathbf{79.82^{\pm0.11}}$ |
| GA, Linear ($\pi_{\text{init}} = 0.75$) | ✓ | $81.01^{\pm0.05}$ | $77.8^{\pm0.3}$ | $21.8^{\pm1.9}$ | $79.80^{\pm0.08}$ |
| GA, Linear ($\pi_{\text{init}} = 0.9$) | ✓ | $80.92^{\pm0.11}$ | $70.6^{\pm8.0}$ | $23.2^{\pm3.7}$ | $79.64^{\pm0.09}$ |

Table 7: Main results for our proposed clipped softmax (CS) and gated attention (GA) applied to ViT-S/16. We report the top-1 accuracy on ImageNet-1K validation set for floating-point baseline and W8A8 quantized model. We also report the maximum $\|\mathbf{x}\|_\infty$ averaged across the validation set, and kurtosis of $\mathbf{x}$ averaged across all layers, where $\mathbf{x}$ is the output of the attention layer.

Detailed results for ViT-S/16 are summarized in Table 7.

After our preliminary experiments on ViT, we noticed that distinct outliers already originate after the patch embeddings. Therefore, we experimented with adding the LayerNorm after the patch

embeddings (which was absent in the model definition, by default). As we can see in Table 6, together with this change, both of our proposed methods greatly dampens the outliers' magnitude, reduces the kurtosis, and yields models with significantly higher quantized performance, which is within 1% of the original FP32 accuracy.

## B.5 The impact of clipped softmax hyperparameters ($\gamma$ and $\zeta$) on ViT

| $\gamma$ | $\zeta$ | FP32 acc. | Max inf norm | W8A8 acc. |
|---|---|---|---|---|
| 0 (= Vanilla) | 1 | $78.80^{\pm0.42}$ | $426^{\pm69}$ | $71.27^{\pm0.88}$ |
| 0 | 1.001 | $78.78^{\pm0.29}$ | $411^{\pm88}$ | $71.24^{\pm0.59}$ |
| 0 | 1.002 | $78.90^{\pm0.17}$ | $420^{\pm47}$ | $70.74^{\pm0.34}$ |
| 0 | 1.004 | $78.80^{\pm0.45}$ | $377^{\pm67}$ | $72.31^{\pm0.06}$ |
| 0 | 1.01 | $78.81^{\pm0.30}$ | $419^{\pm77}$ | $71.35^{\pm0.26}$ |
| $-0.00001$ | 1 | $78.81^{\pm0.21}$ | $432^{\pm76}$ | $69.02^{\pm0.19}$ |
| $-0.0001$ | 1 | $78.81^{\pm0.36}$ | $380^{\pm64}$ | $64.04^{\pm10.8}$ |
| $-0.001$ | 1 | $78.42^{\pm0.63}$ | $282^{\pm105}$ | $68.43^{\pm6.50}$ |
| $-0.003$ | 1 | $\mathbf{78.26^{\pm0.06}}$ | $\mathbf{99^{\pm36}}$ | $\mathbf{76.49^{\pm0.48}}$ |
| $-0.01$ | 1 | $78.10^{\pm0.14}$ | $391^{\pm21}$ | $75.83^{\pm1.12}$ |
| $-0.03$ | 1 | $70.26^{\pm1.46}$ | $197^{\pm2}$ | $65.80^{\pm1.41}$ |
| $-0.001$ | 1.001 | $78.45^{\pm0.53}$ | $283^{\pm82}$ | $65.03^{\pm8.54}$ |
| $-0.003$ | 1.003 | $\mathbf{78.25^{\pm0.14}}$ | $\mathbf{119^{\pm17}}$ | $\mathbf{76.37^{\pm0.45}}$ |

Table 8: The impact of clipped softmax hyperparameters on ViT-S/16.

We investigate the effect of different values of the clipped softmax stretch parameters applied to the vision transformer and present the results in Table 8. To speed up training, for this experiment we trained ViT for 150 epochs instead of the usual 300 epochs. For this experiment, we did not apply LayerNorm after the patch embeddings.

We found similar observations compared to BERT. Specifically, most of the improvement happens when we use $\gamma < 0$ (clipping at zero) whereas using $\zeta > 1$ (clipping at one) yields similar results to the vanilla softmax and combining both $\gamma < 0$ and $\zeta > 1$ yields similar results compared to just clipping at zero.

## B.6 Fine-tuning experiment

| Method | FP16 ppl.$\downarrow$ | Max inf norm | Avg. Kurtosis |
|---|---|---|---|
| Vanilla fine-tuning | 29.46 | 79.3 | 2086 |
| Fine-tuning w/ Gated attention | 29.18 | 50.9 | 665 |

Table 9: OPT-1.3B fine-tuning results with vanilla softmax and gated attention. We report the causal language modeling perplexity (ppl. for short) on the English Wikipedia validation set. We also report the maximum $\|\mathbf{x}\|_\infty$ averaged across the validation set, and kurtosis of $\mathbf{x}$ averaged across all layers, where $\mathbf{x}$ is the output of an attention layer.

One of the drawbacks of our proposed framework is that it requires training from scratch, which could be expensive when applied to very large models. To address this, we explored whether *fine-tuning* using gated attention can still lead to improved performance and decreased outliers for larger models.

We used OPT-1.3B pre-trained checkpoint from HuggingFace and fine-tuned it on Bookcorpus + Wikipedia for 4000 steps with batch size 256, maximum sequence length 512, maximum learning rate $10^{-5}$, and linear LR schedule with 400 warmup steps. We use the same LR for both model parameters and gating module parameters. The rest of hyper-parameters are the same as for our pre-training setup. We adapted our gating approach as follows. We initialized bias as $b_{init} = 0$, which corresponds to the expected initial gating probability output of $\pi_{init} = 0.5$. We multiply the gating probability by 2 so that the expected gate output is 1 and we approximate the attention output of

the vanilla softmax at the start of fine-tuning. We add a small activation regularization term at the output of each FFN to further encourage the reduction in the magnitude of activations, as unlike when training from scratch outliers are already present in the pre-trained model and need to be suppressed.

As we can see from Table 9, fine-tuning with our proposed gated attention results in a better perplexity and also reduced maximum infinity norm and the average kurtosis compared to fine-tuning with vanilla softmax.

### B.7 Low-bit quantization results

| Bitwidths | Weight range estimation | Vanilla | Clipped softmax | Gated attention |
|-----------|------------------------|---------|-----------------|-----------------|
| FP16 | – | $4.49^{\pm 0.01}$ | $4.39^{\pm 0.00}$ | $4.45^{\pm 0.03}$ |
| W8A8 | min-max | $1294^{\pm 1046}$ | $4.52^{\pm 0.01}$ | $4.65^{\pm 0.04}$ |
| W6A8 | min-max | $598^{\pm 254}$ | $4.64^{\pm 0.01}$ | $4.79^{\pm 0.03}$ |
| W6A8 | MSE | $6.49^{\pm 0.38}$ | $4.56^{\pm 0.01}$ | $4.71^{\pm 0.03}$ |
| W4A8 | MSE | $6.52^{\pm 0.02}$ | $4.90^{\pm 0.02}$ | $5.02^{\pm 0.03}$ |
| W6A6 | MSE | $42.8^{\pm 11.7}$ | $6.64^{\pm 0.14}$ | $5.90^{\pm 0.11}$ |

Table 10: A summary of results for our proposed methods applied to BERT-base and quantized to different bitwidthds for weights and activations (using the same PTQ setup as in all previous experiments). We report the masked language modeling perplexity on the English Wikipedia validation set.

Note that our proposed methods are not limited to 8-bit quantization only and in general can be combined with other more advanced quantization and weight compression methods, including [18, 35, 36, 45, 63, 67]. In Table 10, we show the results of our proposed methods applied to BERT-base and quantized to different bitwidths using our simple post-training quantization setup. Unless stated otherwise, for low-bit (<8-bit) weights and activations we use MSE range estimator as recommended by [2, 7] since it gives better results.

As we can see, in all cases both of our methods significantly improve the perplexity compared to the vanilla softmax pre-training. We also notice that generally the performance progressively degrades as we decrease the bitwidths, which is to be expected. Achieving good results with low-bit activation quantization in general is a challenging problem. Further, we notice that the perplexity of the vanilla model significantly improves whenever we consider a low-bit weight quantization with MSE ranges compared to the INT8 case. This can be explained by the fact that using MSE range estimation for weights leads to an implicit clipping of activations (in the same and all subsequent layers in the network), which happen to be of the right amount so that it doesn't hurt the perplexity. We found that by going from W8A8 to W6A8 the average kurtosis is reduced from $3406^{\pm 547}$ to $631^{\pm 94}$ and the maximum infinity norm is reduced from $577^{\pm 80}$ to $158^{\pm 40}$. However, in all cases the resulting model still has significantly larger outliers and a worse performance than both of our proposed methods. Finally, as said before, if achieving good low-bit quantization performance is the goal, it is recommended to combine our methods with more advanced quantization techniques.

## C  Experimental details

### C.1  BERT

**Fine-tuning on MNLI dataset**  We use pre-trained checkpoint BERT-base-uncased (109M parameters) from HuggingFace repository. We follow standard fine-tuning practices from [14] and [65] Each data sequence is tokenized and truncated to the maximum sequence length of 128. Shorter sequences are padded to the same length of 128 using a special [PAD] token. We fine-tune for 3 epochs using Adam [29] with a batch size of 16 and no weight decay. The learning rate is initially set to its maximum value of of $2 \cdot 10^{-5}$ and is linearly decayed to zero by the end of fine-tuning.

**Pre-training from scratch**  We follow closely the pre-training procedure from [14]. We concatenate, tokenize, and split the training set into sequences of length 128 (to speed up training and experimentation, we do not fine-tune on longer sequences of 512). We use the masked language modeling objective with the probability of masking $p = 0.15$. We train with a batch size of 256

sequences for $10^6$ steps, using AdamW optimizer [39] with the maximum learning rate of $10^{-4}$, learning rate warm up over the first $10^4$ steps, following by a linear decay to zero by the end of training. We use L2 weight decay of $0.01$, L2 gradient norm clipping of $1.0$, and dropout probability of $0.1$ on all layers. We also use FP16 mixed-precision from HuggingFace Accelerate library [20].

## C.2 OPT pre-training

To speed up experimentation, we train OPT-125m sized model on the concatenation of Wikipedia and BookCorpus (same as BERT pre-training). We train with a batch size of $48$ and $4$ gradient accumulation steps (which results in the effective batch size of $192$), so that we can perform pre-training on a single A100 80GB GPU. We concatenate, tokenize, and split the training set into sequences of length $512$ and train for $125000$ steps ($500000$ forward passes).

We use the rest of the hyper-parameters and follow pre-training practices from [74] and [65]. We initialize weights using a normal distribution with zero mean and a standard deviation of $0.006$. All bias terms are initialized to zero. We use AdamW optimizer with $(\beta_1, \beta_2) = (0.9, 0.95)$. We use the linear learning rate schedule, warming up from $0$ to the maximum value† of $4 \cdot 10^{-4}$ over the first $2000$ steps, following by a linear decay to zero by the end of training. We use L2 weight decay of $0.1$, L2 gradient norm clipping of $1.0$, and dropout probability of $0.1$ on all layers. We also use FP16 mixed-precision from HuggingFace Accelerate library [20].

## C.3 ViT pre-training

We use the model definition for ViT-S/16 and the training pipeline from PyTorch Image models library [64]. All training is done on resolution $224 \times 224$ and $16 \times 16$ patches. For data augmentation, we use RandAugment [10], Mixup [73], CutMix [70], random image cropping [56], horizontal flip, label smoothing $\varepsilon = 0.1$, color jitter $0.4$, and random (between bilinear and bicubic) interpolation during training.

We train with a batch size of $512$ for $300$ epochs, using AdamW optimizer and the L2 weight decay of $0.03$. We use the cosine learning rate schedule, warming up from $10^{-6}$ to the maximum value of $10^{-3}$ over the first $20$ epochs, followed by a LR decay by a factor of $10$ every $30$ epochs, until it reaches the minimum value of $10^{-5}$.

## C.4 Quantization settings

**Weights** In all cases, we use symmetric uniform quantization of weights. We use min-max weight quantization for all models except the OPT model, for which we found the MSE estimator to perform better in all cases.

**Activations** We adopt *static range estimation* approach, which determines quantization parameters for the network by passing a few batches of calibration data through the model before inference. Specifically, we use a running min-max estimator [32], which uses an exponential moving average of the min and max over multiple batches. In all cases, we use running min-max with $0.9$ momentum over $16$ batches randomly sampled from respective training sets.

For OPT model, we also experiment with using $99.99\%$ and $99.999\%$ percentiles instead of actual min and max. We select the best configuration for each experiment (including baseline), based on the model performance. In almost all cases, we found that setting activation quantization ranges using $99.999\%$ percentiles gives the lowest W8A8 perplexity.

# D Compute cost

We compare the runtime of our proposed methods in Table 11. As we can see, the clipped softmax is only marginally more expensive compared to using the vanilla softmax attention. The gated attention using the linear $G$ adds the compute overhead between $3\%$ and $8\%$, depending on the model. We found that adding weight decay on LayerNorm $\gamma$ for OPT and adding the LayerNorm after the patch embeddings for ViT had a negligible effect on the runtime.

---

†In our experiments, we found this value to perform better compared to the value of $6 \cdot 10^{-4}$ listed in the paper.

| Model | Vanilla | Clipped softmax | Gated attention (Linear / MLP) |
|-------|---------|-----------------|-------------------------------|
| BERT | $92.8^{\pm1.2}$ | $93.6^{\pm0.8}$ | 97.7 / 119.1 |
| OPT | $53.6^{\pm0.4}$ | $54.4^{\pm0.4}$ | 55.7 / 64.7 |
| ViT | $101.8^{\pm0.3}$ | $104.0^{\pm0.7}$ | 110.8 / 122.9 |

Table 11: An overview of the runtime of the proposed methods, compared to the vanilla pre-training, measured in hours on Nvidia-A100 GPUs.

We estimated that the compute cost of producing the main results in the paper is about 320 GPU days (on A100) and the total cost of the project (including preliminary experiments and ablation studies) to be about 1400 GPU days.

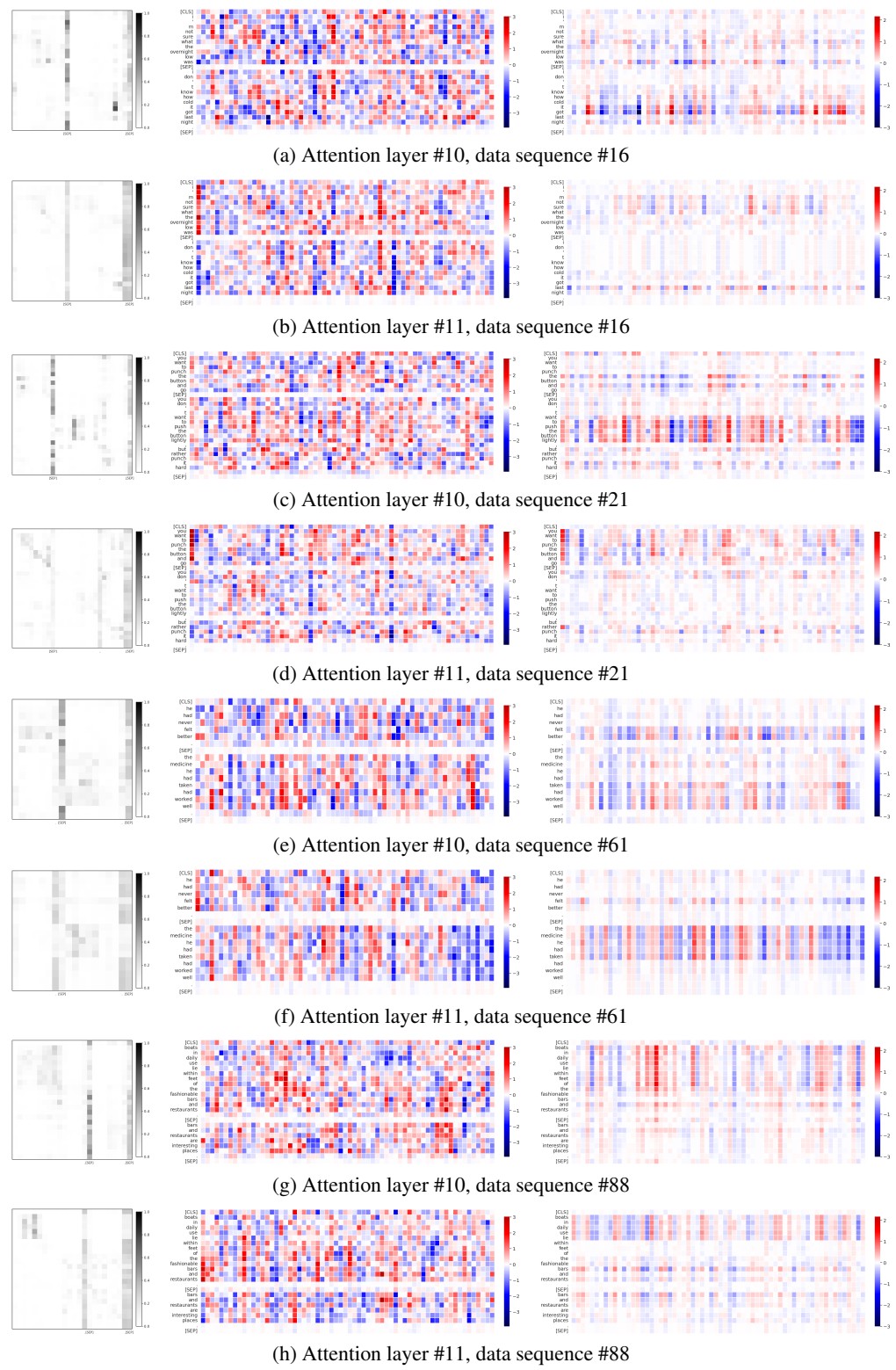

(a) Attention layer #10, data sequence #16

(b) Attention layer #11, data sequence #16

(c) Attention layer #10, data sequence #21

(d) Attention layer #11, data sequence #21

(e) Attention layer #10, data sequence #61

(f) Attention layer #11, data sequence #61

(g) Attention layer #10, data sequence #88

(h) Attention layer #11, data sequence #88

Figure 10: Visualization of the self-attention patterns (attention probabilities, values, and their product in left, middle and right columns, respectively) in attention head #3 ($\leftrightarrow$ channel dim #180) for BERT-base trained with vanilla softmax, computed on several random data sequences from MNLI-m validation set.

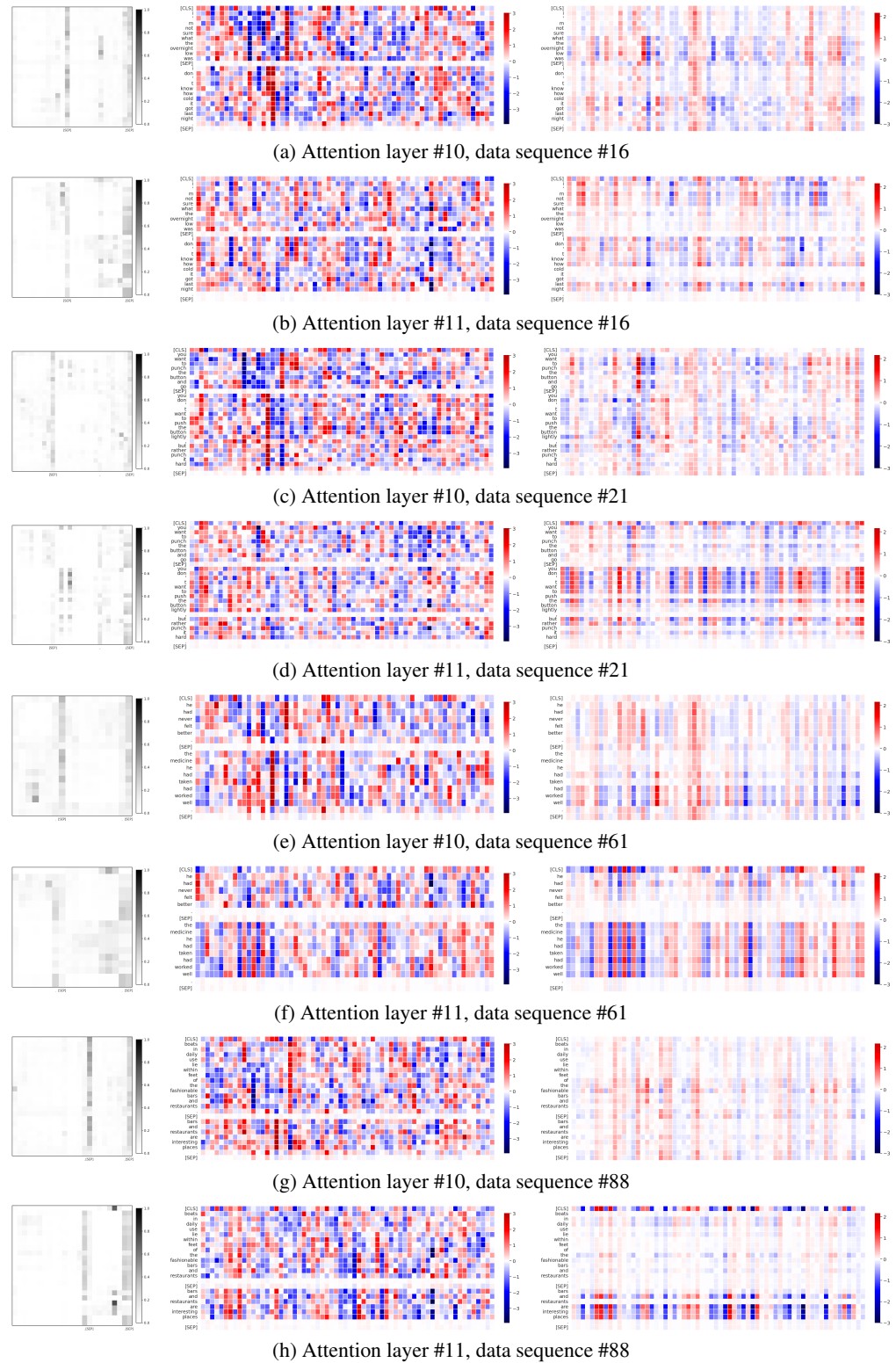

(a) Attention layer #10, data sequence #16

(b) Attention layer #11, data sequence #16

(c) Attention layer #10, data sequence #21

(d) Attention layer #11, data sequence #21

(e) Attention layer #10, data sequence #61

(f) Attention layer #11, data sequence #61

(g) Attention layer #10, data sequence #88

(h) Attention layer #11, data sequence #88

Figure 11: Visualization of the self-attention patterns (attention probabilities, values, and their product in left, middle and right columns, respectively) in attention head #12 ($\leftrightarrow$ channel dim #720) for BERT-base trained with vanilla softmax, computed on several random data sequences from MNLI-m validation set.

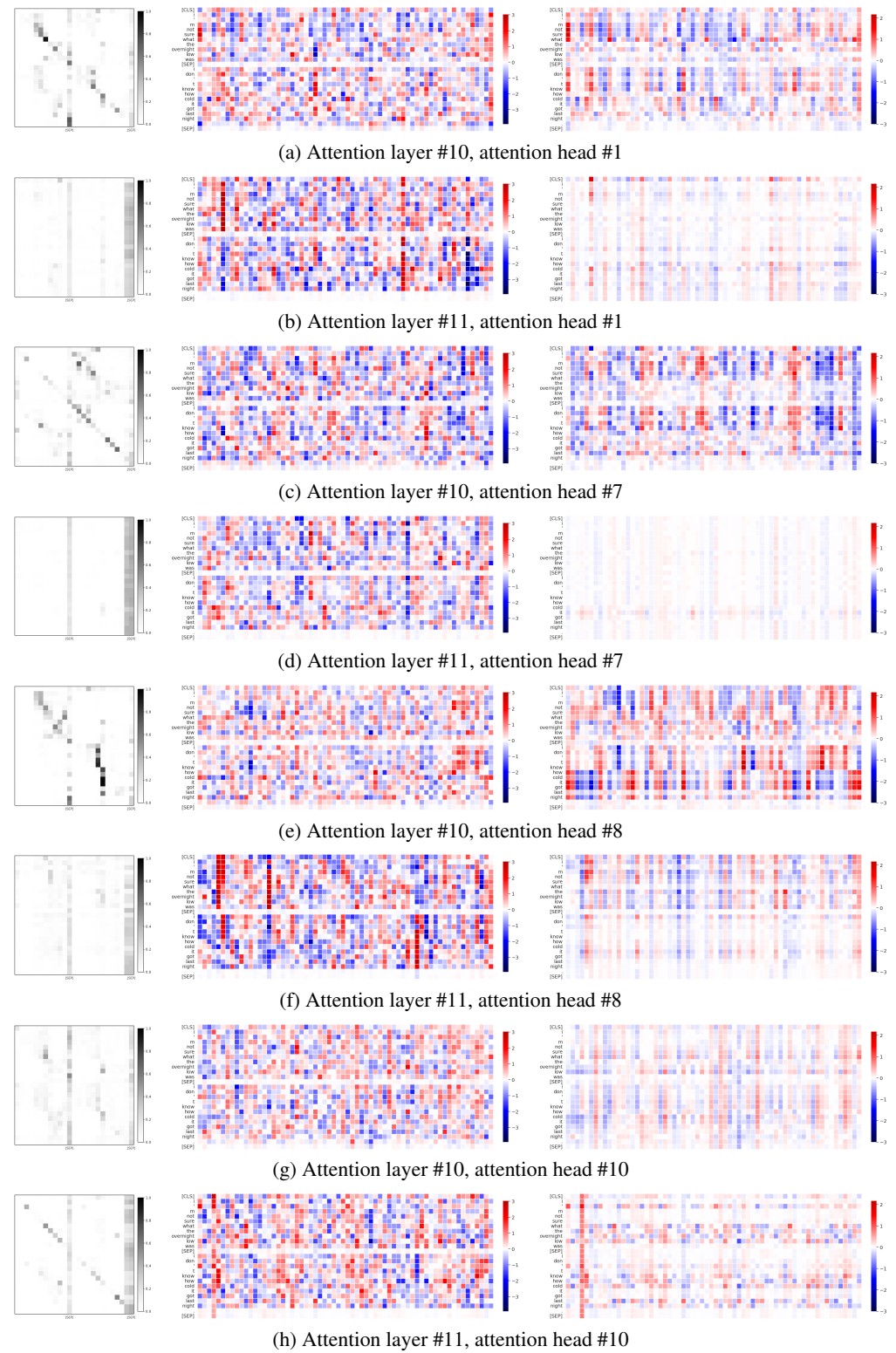

(a) Attention layer #10, attention head #1

(b) Attention layer #11, attention head #1

(c) Attention layer #10, attention head #7

(d) Attention layer #11, attention head #7

(e) Attention layer #10, attention head #8

(f) Attention layer #11, attention head #8

(g) Attention layer #10, attention head #10

(h) Attention layer #11, attention head #10

Figure 12: Visualization of the self-attention patterns (attention probabilities, values, and their product in left, middle and right columns, respectively) in attention heads that are not associated with the strong outliers for BERT-base trained with vanilla softmax, computed on data sequences #16 from MNLI-m validation set.

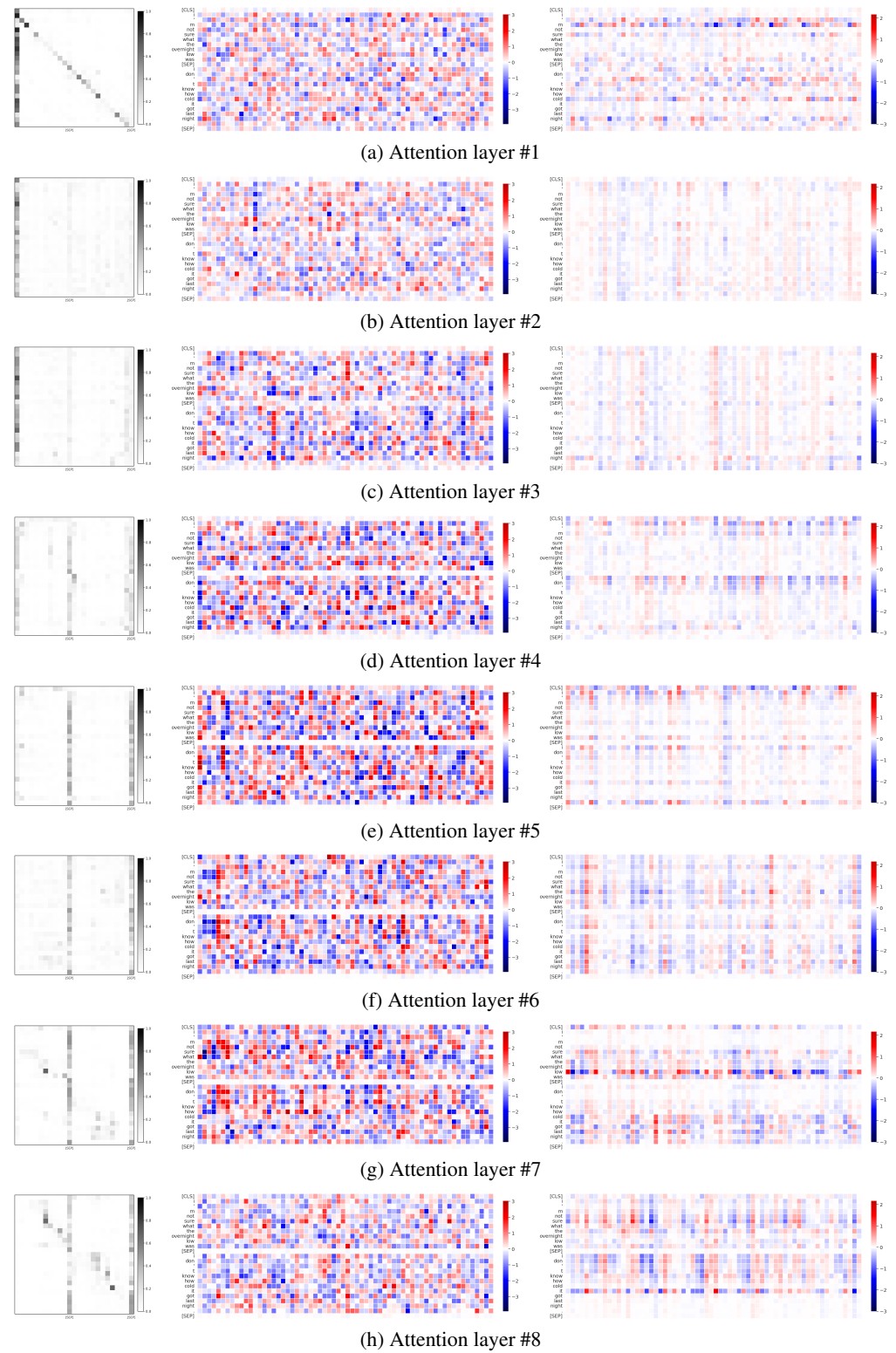

Figure 13: Visualization of the self-attention patterns (attention probabilities, values, and their product in left, middle and right columns, respectively) in attention head #3 ($\leftrightarrow$ channel dim #180) and the first eight layers of BERT-base trained with vanilla softmax, computed on data sequences #16 from MNLI-m validation set.

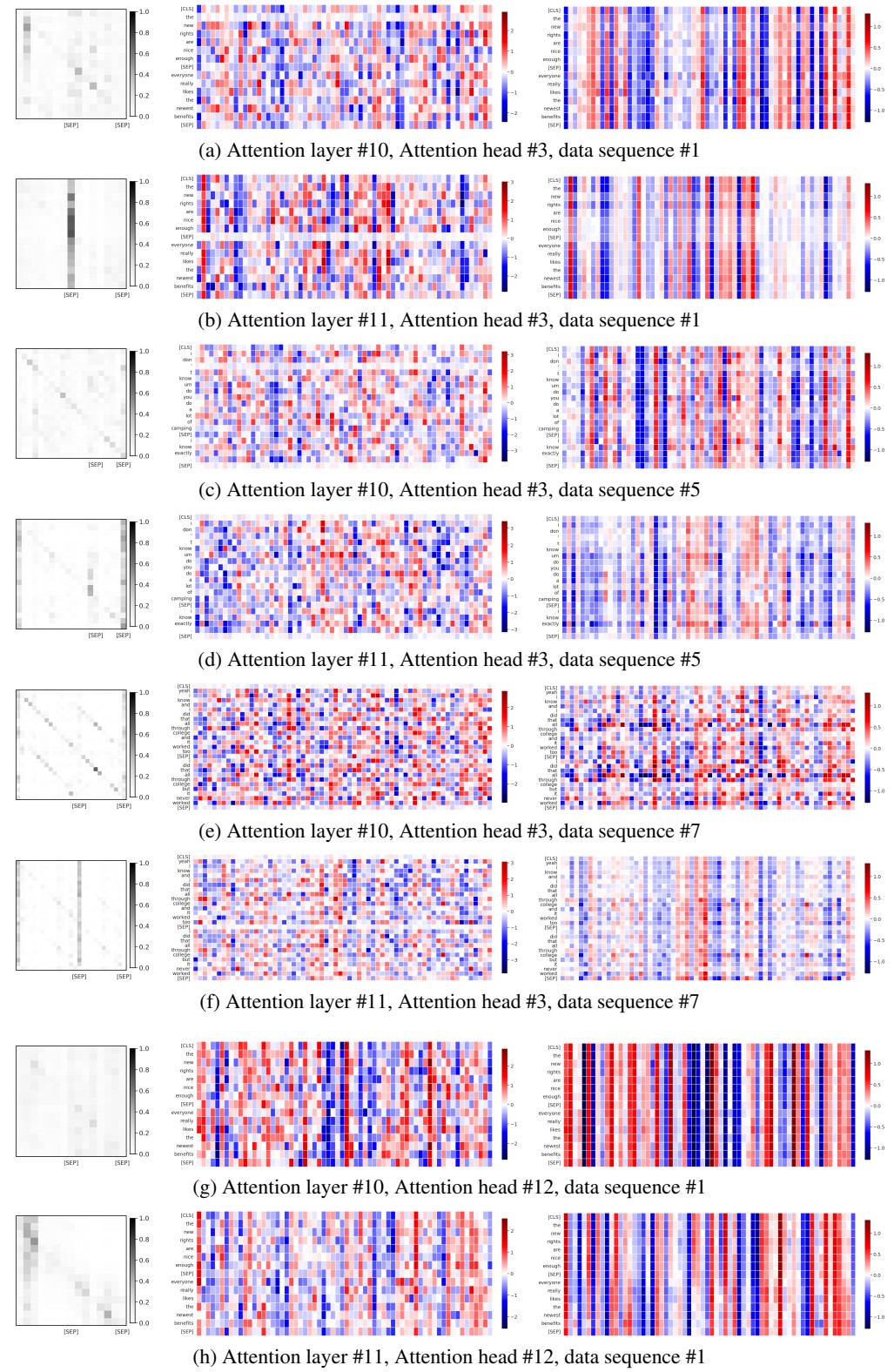

Figure 14: Visualization of the self-attention patterns (attention probabilities, values, and their product in left, middle and right columns, respectively) for BERT-base trained with Clipped softmax, computed on several random data sequences from MNLI-m validation set.

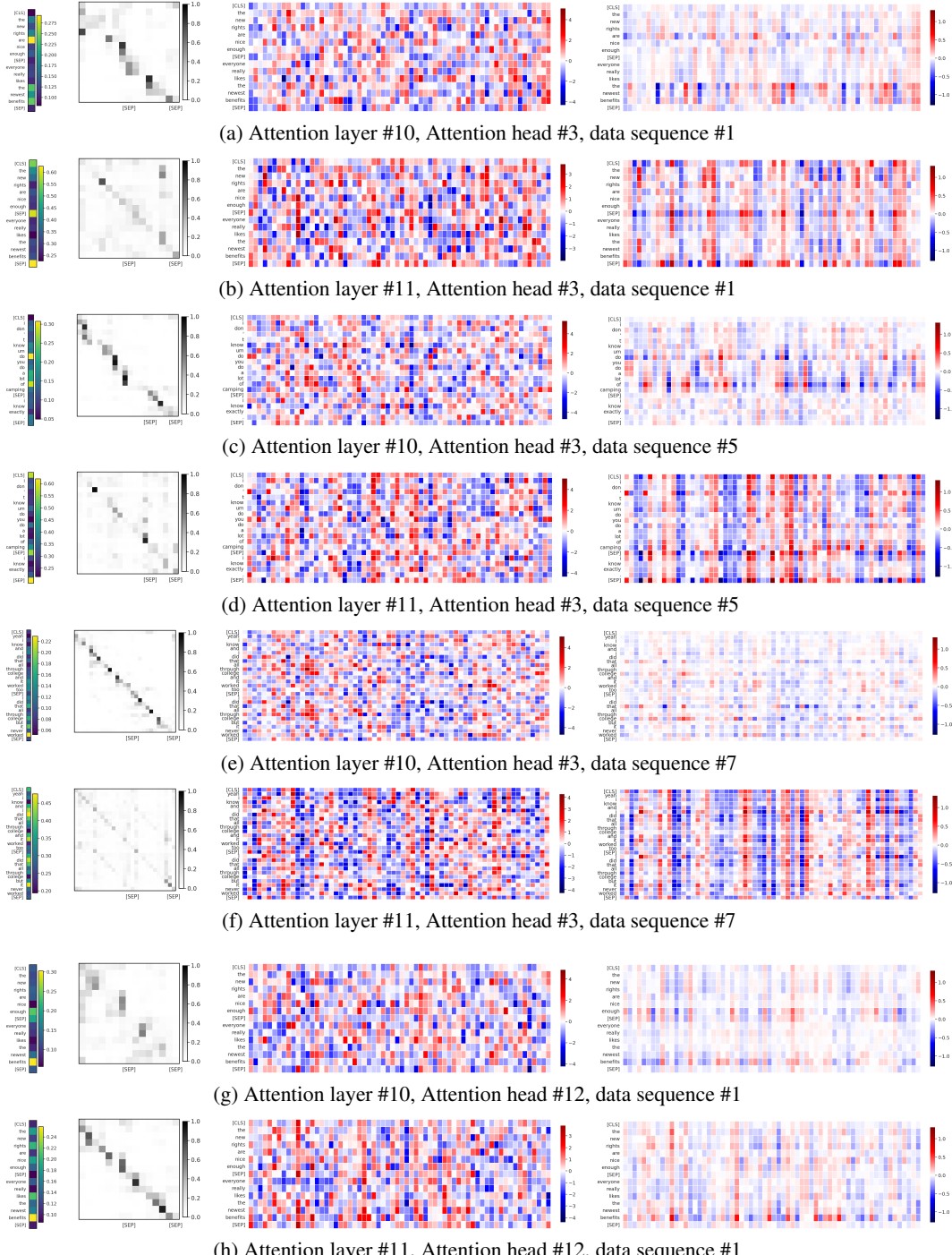

(a) Attention layer #10, Attention head #3, data sequence #1

(b) Attention layer #11, Attention head #3, data sequence #1

(c) Attention layer #10, Attention head #3, data sequence #5

(d) Attention layer #11, Attention head #3, data sequence #5

(e) Attention layer #10, Attention head #3, data sequence #7

(f) Attention layer #11, Attention head #3, data sequence #7

(g) Attention layer #10, Attention head #12, data sequence #1

(h) Attention layer #11, Attention head #12, data sequence #1

Figure 15: Visualization of the self-attention patterns (from left to right: gating probabilities $\boldsymbol{\pi} = \text{sigmoid}\left(\boldsymbol{G}\left(\mathbf{x}\right)\right)$, output of softmax, values, and their combined product) for BERT-base trained with gated attention, computed on several random data sequences from MNLI-m validation set.

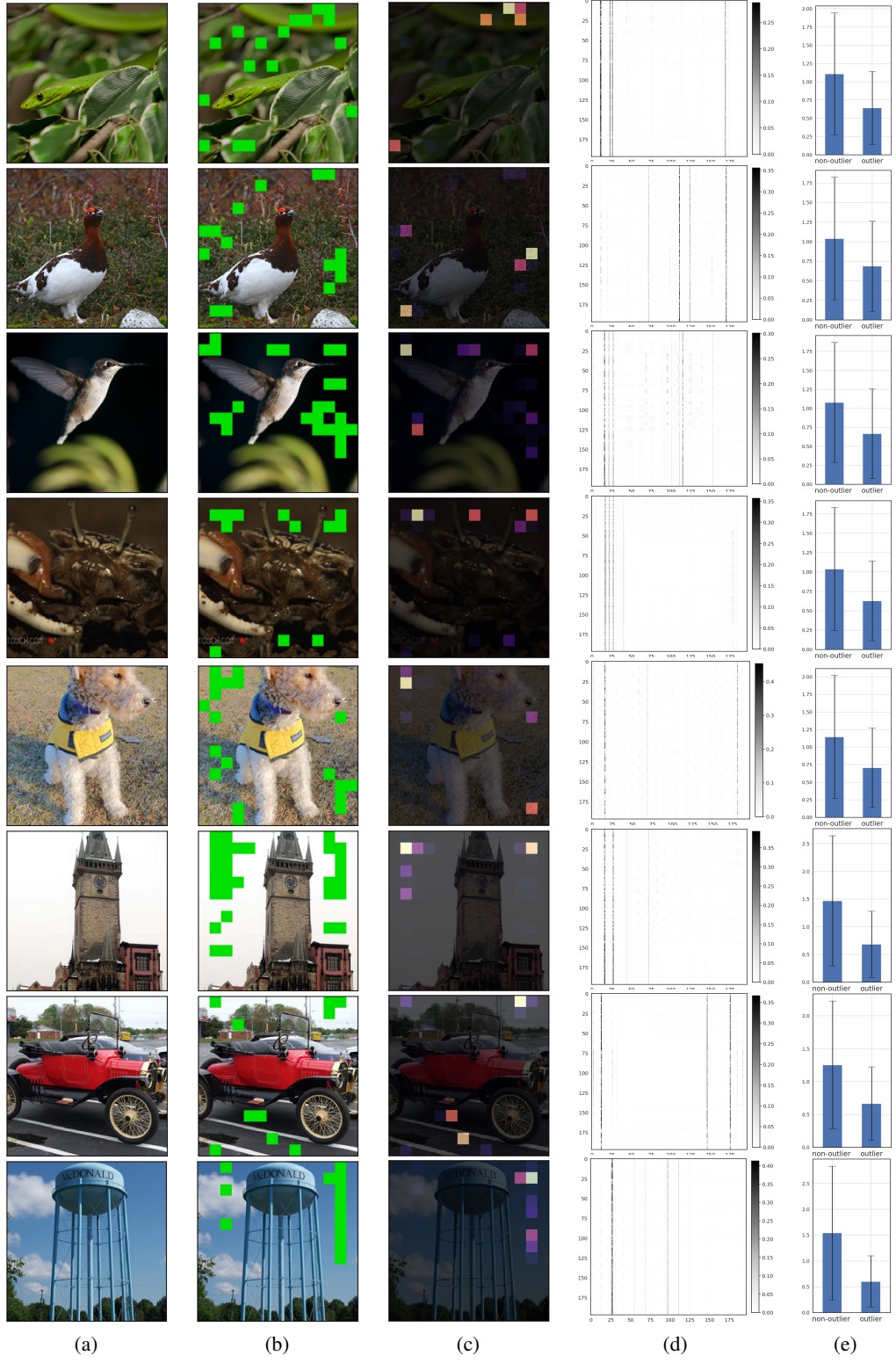

Figure 16: A summary of our outlier analysis for ViT demonstrated on a random subset from ImageNet validation set. (a) An input image. (b) Outliers in the output of layer #10. (c) Cumulative attention weight spent on every patch (matrix of attention probabilities summed over rows) in the attention head #1, in the next layer #11. (d) A corresponding matrix of attention probabilities. (e) An average magnitude of values ($V$) for outlier and non-outlier patches.

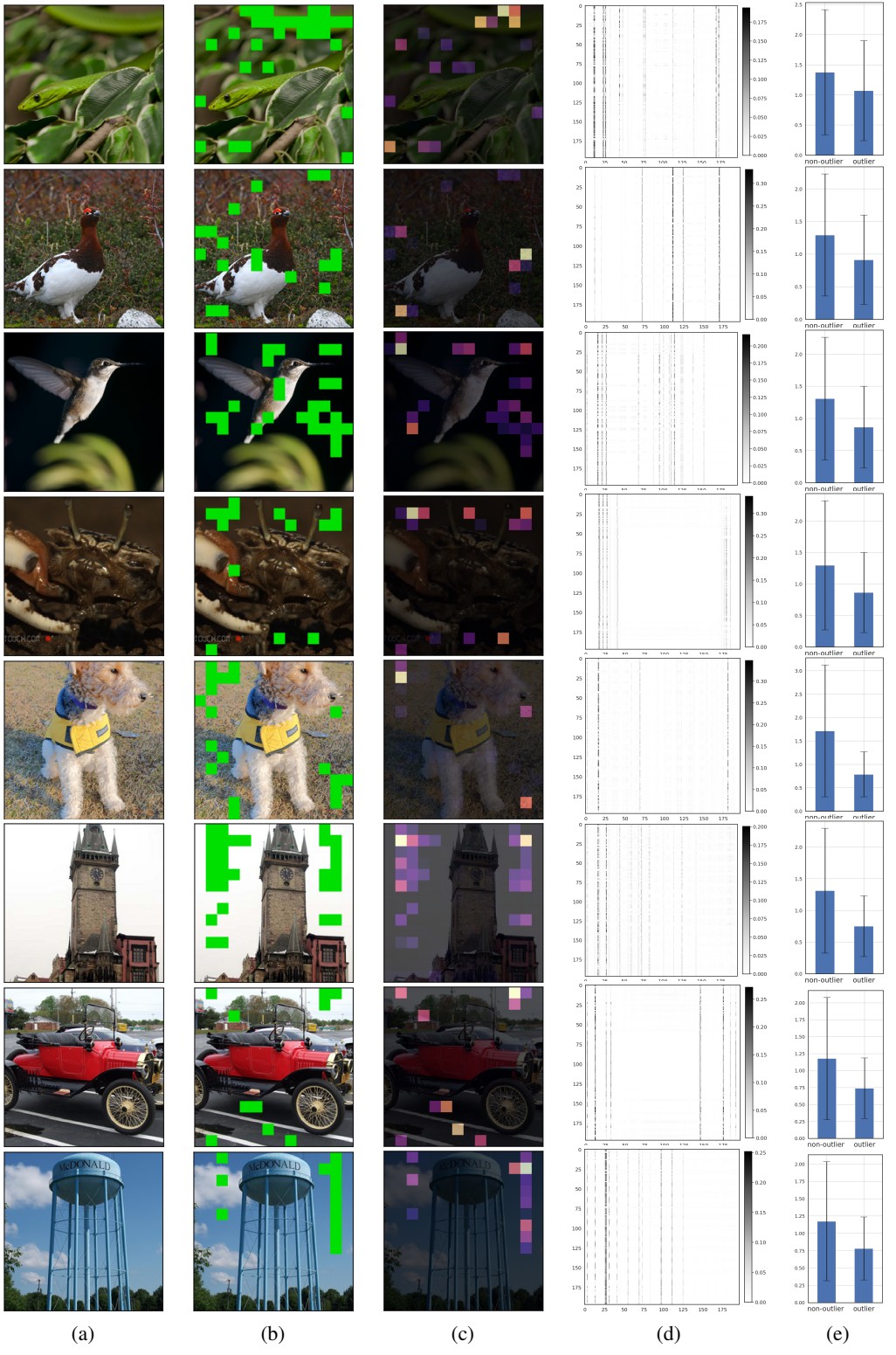

Figure 17: A summary of our outlier analysis for ViT demonstrated on a random subset from ImageNet validation set. (a) An input image. (b) Outliers in the output of layer #11. (c) Cumulative attention weight spent on every patch (matrix of attention probabilities summed over rows) in the attention head #1, in the next layer #12. (d) A corresponding matrix of attention probabilities. (e) An average magnitude of values ($V$) for outlier and non-outlier patches.

