# OpenReview forum: "Quantizable Transformers: Removing Outliers by Helping Attention Heads Do Nothing"
_NeurIPS.cc/2023/Conference — NeurIPS 2023 poster_

### Official Review · Reviewer_Akrm · 2023-06-26

**Soundness:** 4 excellent
**Presentation:** 3 good
**Contribution:** 3 good
**Rating:** 7
**Confidence:** 5

**Summary:**

This paper has thoroughly analyzed the activation outlier problem during quantizing of the transformer model, and further indicates the limitation of existing works. Based on the comprehensive studies, the author proposes two components, namely clipped softmax and gated attention, for regularizing the activation outlier during training. Sufficient experiment results demonstrate the effectiveness and efficiency of the proposed method.


**Strengths:**

$\cdot$ The proposed components, namely clipped softmax and gated attention are simple but reasonable. Further experiments also support their effectiveness.

$\cdot$ The observation is solid and further analysis is practical. The experiments include different structures of the transformer model in various tasks.

$\cdot$ I appreciate the comprehensive studies conducted by the authors. The experiment result of computing cost is convincing.

$\cdot$ This paper is well-organized and clear to understand. And The authors provide detailed information on related methods, making it easy for readers to understand the progress of the field.


**Weaknesses:**

$\cdot$ The quantization performance of the proposed method at a lower bit (e.g. 4W4A) is worth looking forward to.

**Questions:**

$\cdot$ The  avg. kurtosis of OPT model with clipped softmax in tab. 2 is strange, what is the possible reason?

**Limitations:**

$\cdot$ The author has claimed in L290-296. I encourage the authors to finish the experiment in the larger model.

$\cdot$ I encourage the authors to release the code for reproducing.

---

> ### Author Rebuttal · Authors · 2023-08-09
>
> ## Weakness
> See general response to all reviewers.
>
> ## Question
> At the moment, we do not have an explanation of why this is the case but will further investigate this issue.
>
> ## Limitation 1
> See general response to all reviewers.
>
> ## Limitation 2
> We agree that this will greatly improve the reproducibility and accessibility of our work. We intend to release the code with the camera-ready version.

---

> > ### Comment · Reviewer_Akrm · 2023-08-19
> >
> > After reading all reviews and replys, I'd like to keep my score.

---

> > > ### Author Response · Authors · 2023-08-21
> > >
> > > Thanks!

---

### Official Review · Reviewer_NwNc · 2023-07-03

**Soundness:** 3 good
**Presentation:** 4 excellent
**Contribution:** 3 good
**Rating:** 6
**Confidence:** 3

**Summary:**

The paper proposes two architectural modifications that mitigate the activation outlier problem that makes transformers challenging to quantize. Particularly, the authors conduct a comprehensive analysis of the underlying causes of outlier issues in various pre-trained Transformer models across different tasks and datasets. They identify specific architectural limitations that lead to large outliers during the pre-training phase. To tackle this challenge, the authors propose two solutions: 1/ clipped softmax and 2/ gated attention that can effectively address the outlier problem without sacrificing model performance when compared to vanilla Transformer models across various use cases. Furthermore, the suggested modifications result in smaller outlier values and enhance the quantization performance of the Transformer models, making them more robust to quantization.

**Strengths:**

1. The paper presents strong motivations and novel approaches to address outlier issues that distinguish it from prior works. Unlike existing methods that focus on circumventing the impact through post-training techniques, this paper proposes a fundamental training-time solution.
2. The paper proposes a simple yet highly effective methodology, which directly addresses the fundamental issues and observations.

**Weaknesses:**

1. Evaluating the proposed methodology by comparing it solely to the naive application of (vanilla) PTQ on the plain Transformer architecture might be somewhat unfair and not informative enough. Various outlier suppression methods have been proposed [] as post-training solutions, and thus it would be better to include a comparison with these methods. Given that the proposed methodology requires from-scratch training, its practical value will be justified if it outperforms post-training methods that require no additional training.

2. The paper introduces two distinct solutions; however, they are two orthogonal/competing methodologies that tackle the same problem. The paper lacks clarity regarding the specific circumstances under which each solution should be employed. That said, how can we decide which scheme to use when training a new model on a new dataset without having to try both and choose the better one?

3. As also stated in the paper, the evaluations have only been conducted on smaller-scale models (100M parameters). While the proposed architectural and train-time modifications show promising results that match/outperform the vanilla Transformer performance, it is unclear if this trend can persist on a larger scale (hundreds of millions to billions of parameters). It would be more informative if the authors can provide across different model scale regimes. Moreover, since the proposed solution is a pre-training methodology, it requires training from scratch and additional risk/cost related to extra hyperparameter tuning, which can potentially limit their practical application.


[1] Wei, Xiuying, et al. "Outlier suppression: Pushing the limit of low-bit transformer language models." Advances in Neural Information Processing Systems 35 (2022): 17402-17414.

[2] Xiao, Guangxuan, et al. "Smoothquant: Accurate and efficient post-training quantization for large language models." arXiv preprint arXiv:2211.10438 (2022).

**Questions:**

N/A

**Limitations:**

Please see the weakness section

---

> ### Author Rebuttal · Authors · 2023-08-09
>
> ## Weakness 1
> Indeed, in most of real-world applications the vanilla PTQ might not be good enough.
>
> At the same time, our proposed methodology is complementary and can be combined with most of more advanced PTQ and weight compression techniques ([1],[2] as well as [3-6] and many more).
>
> Our main motivation of choosing this PTQ baseline was to show how effective our methods are at recovering the quantized network performance even with such a naive and minimal effort PTQ pipeline.
>
> Please also note that, as stated in the main text, in most of these works, they keep certain parts of the network (often including the problematic residual connections after FFN, LayerNorm etc.) in FP16. On the contrary, we quantize all weigths and activations, including the problematic input, output and residual connections in FFN.
>
> References:
> * [3] Frantar, E., Ashkboos, S., Hoefler, T., & Alistarh, D. (2022). Gptq: Accurate post-training quantization for generative pre-trained transformers. arXiv preprint arXiv:2210.17323.
> * [4] Lin, J., Tang, J., Tang, H., Yang, S., Dang, X., & Han, S. (2023). AWQ: Activation-aware Weight Quantization for LLM Compression and Acceleration. arXiv preprint arXiv:2306.00978.
> * [5] Nagel, M., Amjad, R. A., Van Baalen, M., Louizos, C., & Blankevoort, T. (2020, November). Up or down? adaptive rounding for post-training quantization. In International Conference on Machine Learning (pp. 7197-7206). PMLR.
> * [6] Li, Y., Gong, R., Tan, X., Yang, Y., Hu, P., Zhang, Q., ... & Gu, S. (2021). Brecq: Pushing the limit of post-training quantization by block reconstruction. arXiv preprint arXiv:2102.05426.
>
> ## Weakness 2
> This is a good point!
> * Given the fact that clipped softmax did not work for OPT, for the moment we would recommend starting with gated attention (that worked consistently well on all tested models so far) approach but also exploring clipped softmax for new models.
> * On top of that, we expect to further bring more clarity to this question by exploring bigger-scale models.
>
> ## Weakness 3
> We agree on risks regarding the hyper parameter tuning and with all the other points.
> Please, see general response to all reviewers.

---

> > ### Comment · Reviewer_NwNc · 2023-08-20
> >
> > I appreciate the authors for their thoughtful response. I will keep my rating and stand for acceptance.

---

> > > ### Author Response · Authors · 2023-08-21
> > >
> > > Thanks!

---

### Official Review · Reviewer_Vxgc · 2023-07-04

**Soundness:** 3 good
**Presentation:** 3 good
**Contribution:** 3 good
**Rating:** 6
**Confidence:** 4

**Summary:**

In this paper, the authors show that strong outliers are related to very specific behavior of attention heads that try to learn a “no-op” or just a partial update of the residual. To reduce outliers, the authors propose two simple (independent) modifications to the attention mechanism. This enables them to quantize transformers to full INT8 quantization of the activations without any additional effort.

**Strengths:**

1. The authors provide a detailed analysis of the outliers in transformers, which well clarifies their research motivation and provides insights.
2. The analysis of "strong outliers are related to attention heads try to learn no-op" is interesting.

**Weaknesses:**

1. The authors do not give detailed data distribution after training with the proposed method. I would expect a reduction in the magnitude and number of outliers than in the performance of INT8 quantization that validates the main target of the proposed technique.
2. The authors mention that it is limited to small models only (125M, which is too small), but quantizing small models with less than 2.7B parameters to INT8 is less challenging compared with large models  [1]. There exists concern is the generalization ability of the proposed method on large-scale models. Analysis or experiments that demonstrate the effectiveness of proposed technique on larger scale model is recommended.

[1] LLM.int8(): 8-bit Matrix Multiplication for Transformers at Scale. NIPS'22

**Questions:**

1. Could the proposed method be applied to lower bitwidths beyond INT8?  (e.g., 4/6 bit)
2. Does the proposed method require a full retraining of the FP model? When applied to larger models, the training cost could be excessive.

**Limitations:**

The authors do not include the limitations and potential negative societal impact in their paper.

---

> ### Author Rebuttal · Authors · 2023-08-09
>
> ## Weakness 1
> Please note, that in all result tables we report next to the floating-point and quantized perplexity/accuracy also two metrics that quantify the magnitude and the frequency of the outliers:
> 1) Maximum infinity norm - measures the magnitude of the outliers (by definition).
> 2) Kurtosis - the (empirical) fourth standardized moment - measures the tailedness of a distribution, which is directly related to the frequency of the outliers in the distribution (the heavier the tails of the distribution - the higher the kurtosis and vice versa).
>
> And as we can see from Table 2, in almost all cases, both of our proposed techniques significantly reduce both the maximum infinity norm (the magnitude) and the kurtosis (the frequency) of the outliers.
>
> ## Weakness 2
> See general response to all reviewers.
>
> ## Question 1
> See general response to all reviewers.
>
> ## Question 2
> * Indeed, the idea is that the model is pre-trained from scratch using one of our proposed techniques instead of the vanilla softmax which result in a way more quantization-friendly model. We hope that gated attention and/or clipped softmax will be picked up for future LLM generation such that models are readily quantizable.
> * However, we agree that this could lead to excessive training costs when applied to very large models.
> * To address this, we explored whether fine-tuning using Gated attention can still lead to improved performance and decreased outliers for larger models.
> * Setup:
> 	- we used OPT-1.3B pre-trained checkpoint from HuggingFace and fine-tuned it on Bookcorpus + Wikipedia for 4000 steps with batch size 256, maximum sequence length 512, LR = 1e-5, and linear LR schedule with 400 warmup steps (we use the same LR for both model parameters and gating module parameters) and the rest of hyper-parameters are the same as for our pre-training setup.
> 	- we adapted our gating approach as follows: we used b_init = 0 which results in the expected initial gating probability output of pi_init = 0.5. We multiply the gating probability by 2.0 so that at initialization this value is 1 and approximately resembles the model with vanilla softmax at the start of fine-tuning.
> 	- we add a small activation regularization term (at the output of each FFN) to further encourage the reduction in the magnitude of activations (as unlike when training from scratch outliers are already present in the pretrained model and need to be suppressed).
> * We show results in Table 2 in the attached PDF. As we can see, fine-tuning with our proposed Gated attention results in a better perplexity and also reduced maximum infinity norm and the average kurtosis compared to fine-tuning with vanilla softmax.
>
> ## Limitations
> Please note that in the discussion (Section 6), we mention some limitations and a few points regarding the potential societal impact. In case we missed any limitation or potential negative societal impact, we are happy to update the section based on the reviewer’s suggestion(s).

---

> > ### Comment · Reviewer_Vxgc · 2023-08-18
> > **Thanks for the authors' response**
> >
> > Thanks for the authors' response. I keep my rating that recommends accept.

---

> > > ### Author Response · Authors · 2023-08-18
> > >
> > > Thanks!

---

### Official Review · Reviewer_sWJn · 2023-07-07

**Soundness:** 3 good
**Presentation:** 3 good
**Contribution:** 2 fair
**Rating:** 6
**Confidence:** 5

**Summary:**

This paper analyzes the activation outlier problem. It is shown that the outliers are related to the behavior of transformer networks trying to learn not to update residuals (no-op). To achieve the exact zeros needed in the attention matrix for a no-update, the input to the softmax is pushed to be larger and larger. To solve this problem, the authors propose clipped softmax method and gated attention method, which can introduce exact zeros for softmax, thus the outliers can be removed.

**Strengths:**

The paper explains why outliers exist in transformers. The analysis and visualization are convincing.

Two independent methods are proposed (clipped softmax and gated attention) to solve the outlier problem of transformers, so that models can be quantized easily.

Experiments on language models (BERT, OPT) and vision transformers are provided.

**Weaknesses:**

No results on large scale models and datasets are provided.

**Questions:**

How to explain the clipped softmax method no better than vanilla network on OPT?

Can you provide the visualization (Figure 2) after using the proposed methods?

**Limitations:**

Yes

---

> ### Author Rebuttal · Authors · 2023-08-09
>
> ## Weakness 1
> See general response to all reviewers.
>
> ## Question 1
> At the moment, we do not have an explanation of why this is the case but will further investigate this issue.
>
> ## Question 2
> Thanks for the suggestion. We investigated this and included the visualization, please see Figure 1 in the attached PDF.
>
> As we can see, both methods can represent a partial/soft no-op behavior, but in case of our methods this does not require strong outliers elsewhere in the network:
> * In the case of clipped softmax, the attention probabilities are generally more diffused and smaller in magnitude (which comes from the clipping).
> * In the case of gated attention, the output of softmax is also quite different since the update of the hidden representation is now further modulated by the gating probabilities.
>
> Note that we found similar patterns in multiple attention heads, but the exact head indices where we observed such patterns depend on random initialization.

---

> > ### Comment · Reviewer_sWJn · 2023-08-17
> > **Response to rebuttal**
> >
> > Thanks for the reply. Figure 1 in the attached PDF uses different head and sequence for clipped softmax and vanilla softmax. I suggest to use the same setting for visualization, even if the visualization results are not as expected, it will help understanding the method.

---

> > > ### Author Response · Authors · 2023-08-18
> > >
> > > Thanks for the comment.
> > >
> > > Initially, we looked at and analyzed the same head and same sequence but decided to showcase a different head and a sequence with a similar attention pattern of soft no-update/partial update (we would show both if not the 1-page limit for the attached PDF).
> > >
> > > Such a pattern was not as pronounced as was the case presented in Figure 2 in the main text, but it's also not surprising (because of different random initialization & training dynamics it doesn't have to be the same exact head).
> > >
> > > However, we agree that including the same head & sequence is also valuable and we include both and possibly a few more for the camera-ready version.

---

### Author Rebuttal · Authors · 2023-08-09

We thank all reviewers for their thoughtful and positive feedback!

We are encouraged they found our work is well-organized and easy to follow (Akrm), has comprehensive experiments (sWjn, Akrm), solid and insightful analysis (sWjn, Vxgc ,Akrm), and presents a simple yet effective methodology, which directly addresses the observations and fundamental issues of outliers (NwNc, Akrm).

One common concern (all reviewers) is the question of the scalability of our methods to larger models, including LLMs.
* First, we would like to clarify that our methods are not limited to the smaller-scale models and we expect them to translate to larger models/LLMs. We did not include the results for larger models purely because of compute constraints.
* We acknowledge that this is a very important question and that showing that our methods remain effective at a larger scale will greatly improve the likelihood of the adoption of our methods.
* We intend to include the experiments on bigger models with the camera-ready version (up to 1.5B model size).

Several reviewers (Vxgc, Akrm) have also asked if the proposed method be applied to lower bitwidths beyond INT8 (e.g., 4/6 bits)?
* Indeed, the proposed methods can be applied to lower bitwidths and can also be combined with other more advanced PTQ techniques.
* Please, see Table 1 in the attached PDF the results for our proposed methods applied to BERT-base and quantized to different bitwidths using our simple PTQ setup. Unless stated otherwise, for low-bit (<8-bit) weights and activations we use MSE range estimation as recommended by [1,2] since it gives better results.
* As we can see, in all cases both of our methods significantly improve the perplexity compared to the vanilla softmax pre-training.
* We also notice that generally the performance progressively degrades as we decrease the bitwidths, which is as expected. Achieving good results with low-bit activation quantization in general is a challenging problem to this day.
* Finally, we notice that the perplexity of the vanilla model significantly improves whenever we consider a low-bit weight quantization with MSE ranges compared to the INT8 case. This can be explained by the fact that using MSE range estimation for weights leads to an implicit clipping of activations (in the same and all subsequent layers in the network), which happen to be of the right amount so that it doesn't hurt the perplexity. We found that by going from W8A8 to W6A8 the average kurtosis is reduced from 3406±547 to 631±94 and the maximum infinity norm is reduced from 577±80 to 158±40. However, in all cases the resulting model still has significantly larger outliers and a worse performance than both of our proposed methods.

References:
* [1] Ron Banner, Yury Nahshan, Elad Hoffer, and Daniel Soudry. 2018. Post-training 4-bit quantization of convolution networks for rapid-deployment. arXiv preprint arXiv:1810.05723
* [2] Yoni Choukroun, Eli Kravchik, Fan Yang, and Pavel Kisilev. 2019. Low-bit quantization of neural networks for efficient inference. In ICCV Workshops, pages 3009–3018.

---

### Decision · Program_Chairs · 2023-09-21

**Decision:**

Accept (poster)

**Comment:**

All reviewers agree that this paper proposes an interesting hypothesis for the outlier problem in transformers. The paper presents enough evidence and further evaluates two simple approaches to mitigate the solution. Main concerns were about the scale of the experiments, and in-consistency in performance of the proposed approaches. Authors provided further experiments in the response and promised to include more in the final version, and acknowledged the inconsistency in some of the approaches, which satisfied reviewer's concerns. Overall, the paper makes clean contribution to an important problem. All reviewers agree on accepting this paper, and I am happy to suggest acceptance.